# Atomic sawtooth-like metal films for vdW-layered single-crystal growth

Hayoung Ko [1,8], Soo Ho Choi[1,2,8], Yunjae Park [3,8], Seungjin Lee [1], Chang Seok Oh[1], Sung Youb Kim[3,4], Young Hee Lee [1,2] ✉, Soo Min Kim[5] ✉, Feng Ding [6,7] ✉ & Ki Kang Kim [1,2] ✉

Atomic sawtooth surfaces have emerged as a versatile platform for growth of single-crystal van der Waals layered materials. However, the mechanism governing the formation of single-crystal atomic sawtooth metal (copper or gold) films on hard substrates (tungsten or molybdenum) remains a puzzle. In this study, we aim to elucidate the formation mechanism of atomic sawtooth metal films during melting–solidification process. Utilizing molecular dynamics, we unveil that the solidification of the liquid copper initiates at a high-index tungsten facet with higher interfacial energy. Subsequent tungsten facets follow energetically favourable pathways of forming single-crystal atomic sawtooth copper film during the solidification process near melting temperature. Formation of atomic sawtooth copper film is guaranteed with a film thickness exceeding the grain size of polycrystalline tungsten substrate. We further demonstrate the successful growth of centimeter-scale single-crystal monolayer hexagonal boron nitride films on atomic sawtooth copper films and explore their potential as efficient oxygen barrier.

Two-dimensional (2D) van der Waals (vdW) layered materials exhibit exotic physical and chemical properties such as high carrier mobility, strong spin-orbit coupling, room-temperature ferromagnetism, superconductivity, and high quantum yields[1–6]. With these unique characteristics, they have considerable potential for various high-performance device applications, including optoelectronics, spintronics, and twistronics[6–8]. The availability of wafer-scale single-crystal (SC) films is crucial to fully exploit the intrinsic properties of these materials in such devices[9]. In recent years, significant advancements have been achieved in the growth of wafer-scale SC 2D van der Waals (vdW) films using chemical vapor deposition (CVD) techniques[10–22]. Achieving wafer-scale SC 2D vdW films requires the coherent alignment of 2D grains during the growth process[10]. Two strategies

have been proposed to achieve coherent alignment: self-collimation of grains on a liquid substrate and epitaxial growth on an SC solid substrate[11–22]. The former approach involves the movement and rotation of circular hexagonal boron nitride (hBN) grains on a liquid Au surface to ensure alignment in a single direction via electrostatic Coulomb interaction between the grains[10]. The aligned circular grains merge to form an SC hBN film; however, only monolayer hBN films have been demonstrated thus far. The latter approach requires a specific SC-growth substrate, featuring the vicinal surface is an ideal growth platform for SC 2D vdW materials[20–23]. The importance of vicinal surfaces for SC growth has also been proposed through studies using density functional theory (DFT) and molecular dynamics[11].

---

[1]Department of Energy Science, Sungkyunkwan University (SKKU), Suwon 16419, South Korea. [2]Center for Integrated Nanostructure Physics (CINAP), Institute for Basic Science (IBS), Sungkyunkwan University (SKKU), Suwon 16419, South Korea. [3]Graduate School of Carbon Neutrality, Ulsan National Institute of Science and Technology (UNIST), Ulsan 44919, South Korea. [4]Department of Mechanical Engineering, Ulsan National Institute of Science and Technology (UNIST), Ulsan 44919, South Korea. [5]Department of Chemistry, Sookmyung Women's University, Seoul 14072, South Korea. [6]Department of Materials Science and Engineering, Ulsan National Institute of Science and Technology (UNIST), Ulsan 44919, South Korea. [7]Shenzhen Institute of Advanced Technology, Chinese Academy of Science, Shenzhen, China. [8]These authors contributed equally: Hayoung Ko, Soo Ho Choi, Yunjae Park. ✉e-mail: leeyoung@skku.edu; soominkim@sookmyung.ac.kr; f.ding@siat.ac.cn; kikangkim@skku.edu

The SC vicinal surfaces of copper (Cu) (110)/(111) and gold (Au) (111), high-index Cu, miscut sapphire, and atomic sawtooth (AS) surfaces of Cu/Au, have been employed to grow SC monolayered vdW 2D materials[12–19]. The AS surface consists of a mixture of various step-terrace surfaces, referred to as vicinal surfaces or high-index surfaces, across its corrugated macroscopic surface. Particularly, its terraces exhibit the same low Miller indices of (111), (110), or (100) on a wafer scale[18,19]. Notably, Cu serves as an economical substrate for the large-scale production of graphene and hBN. However, establishing SC vicinal and high-index Cu surfaces via abnormal grain growth for graphene and hBN films requires an extended thermal-annealing procedure of over 24 h. In addition, the preparation of miscut sapphire for SC transition-metal dichalcogenide (TMD) films requires meticulous cutting techniques. Furthermore, the significant cost of sapphire production restricts its mass production[12–17,24]. Upon solidifying molten Cu (or Au) on a polycrystalline (PC) W foil, wafer-scale AS SC metal films are formed, which serve as substrates for the synthesis of coherently aligned 2D grains[18,19]. This MS procedure requires only a few hours to produce large-area SC AS substrates and eliminates the need for complex cutting or abnormal grain-growth processes. Previous studies have demonstrated the SC growth of graphene, TMDs, and TMD heterostructures on AS metal films[18,19]. However, the mechanism underlying the formation of AS metal films on PC metal substrates remains unclear. Thus, a comprehensive understanding of this mechanism is essential to further expand the SC-growth platform of 2D materials and vdW heterostructures.

In this study, we investigate the mechanisms underlying the formation of SC AS metal films, specifically Cu and Au, on PC metal substrates of either W or Mo. We fabricate SC AS Cu films on PC W substrates using an MS procedure. Electron-backscatter diffraction (EBSD) and cross-sectional transmission electron microscopy (TEM) are employed to confirm the formation of these films. To understand the intricacies of the MS process, we perform molecular-dynamics (MD) simulations to ascertain the energetic relationships at the interfaces between the metal overlayers and substrates. Our observations reveal that the epitaxial solidification of liquid Cu (or Au) was initiated at high-index W facets with greater interface energy. As this solidification occurred, it extends to the W facets with lower interface energies to minimize the overall energy, resulting in the creation of SC AS Cu films. Interestingly, we note that epitaxial solidification started at a randomly chosen W facet, resulting in a random Miller index for the SC AS Cu films. Analogous behaviors are also observed for Au layers on PC W or Mo substrates. To demonstrate the viability of the concept, we grow SC hBN films on SC AS Cu films to further assess their potential in oxygen-barrier applications.

## Results and discussion

### Single-crystal atomic sawtooth metal film via melting–solidification method

The MS method was used to prepare wafer-scale SC AS metal (Cu or Au) films (Fig. 1a). The procedure involved stacking pre-cleaned PC metal foils onto a PC W foil that served as the substrate. The stack was placed at the center of a furnace. The Cu metal foils were subsequently heated beyond their melting points of ~1085 °C. The molten Cu was then cooled to 1050 °C at a cooling rate of <0.2 °C/min, resulting in the formation of wafer-scale SC AS metal films. The Miller indices of the metal surfaces of annealed PC Cu and W foils at ~1050 °C, as well as that of the SC AS metal film, were evaluated using EBSD measurements. The EBSD mapping images of the W and Cu foils revealed randomly oriented small grains with areas of ~4.83 (±1.27) and ~11,305.82

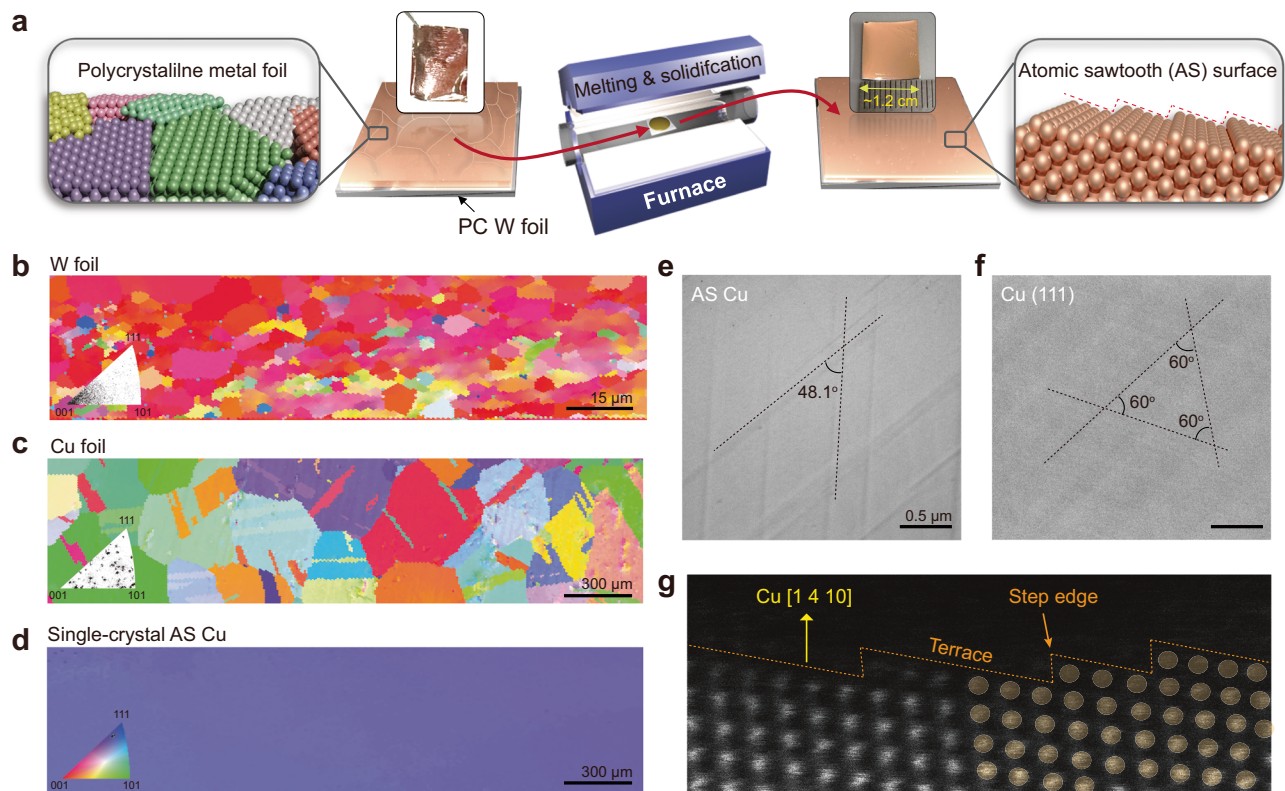

**Fig. 1 | Large-scale single-crystal (SC) atomic sawtooth (AS) metal film on polycrystalline (PC) substrates. a** Schematic representation of the melting–solidification (MS) process used to form AS metal films on PC W foils. The insets in (**a**) show photographs of Cu substrates before and after the MS process. Electron backscatter diffraction (EBSD) images of (**b**) annealed PC W foil, (**c**), Cu foil, and (**d**), SC AS Cu film, with an inverse pole figure (IPF) map. **e, f** Representative scanning electron microscopy (SEM) images of SC AS Cu and Cu (111). **g** Cross-sectional annular dark-field scanning transmission electron microscopy image of SC AS Cu film.

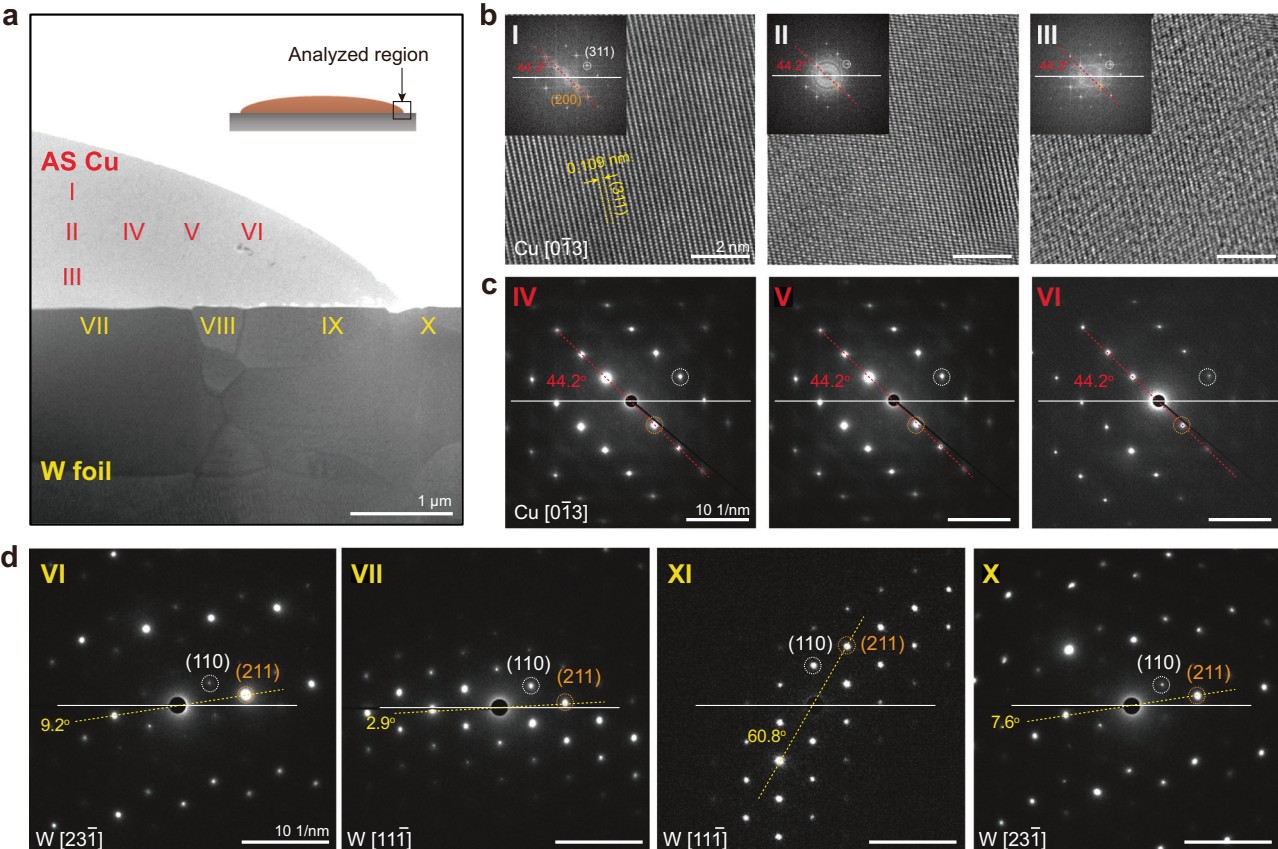

**Fig. 2 | Cross-sectional TEM analysis of an SC AS Cu film and PC W foil. a** Low-magnification cross-sectional annular bright field scanning transmission electron microscopy image of an SC AS Cu film on a W foil. **b** High-resolution transmission electron microscopy (HR-TEM) images of the SC AS Cu, which were obtained from regions I–III in (**a**). Insets in (**b**) show corresponding fast-Fourier-transform (FFT) images. **c** Selective area electron diffraction (SAED) patterns of the SC AS Cu, measured at regions IV–VI in (**a**). **d** SAED patterns of the PC W foil obtained at regions VII–X in (**a**).

(±1191.74) μm² (Figs. 1b, c, and S1) including the low and high Miller indices, respectively. In contrast, the EBSD mapping image of the SC AS Cu film shows a near-uniform color contrast over a centimeter scale, implying the presence of a prevalent SC AS surface across the entire region (Figs. 1d and S2). Inverse pole figures (IPFs) were distinctly delineated between the PC and SC surfaces (insets of Fig. 1b–d). It is noted that while the optimal cooling rate (<0.2 °C/min) resulted in the formation of a SC AS metal film, faster cooling rates (3.3 and 16 °C/s) led to the formation of PC AS metal film (Fig. S3). This observation may be attributed to the initiation of several nuclei simultaneously at multiple points during the rapid cooling process. Notably, the angle between the slip lines in the representative scanning electron microscopy (SEM) image of the SC AS Cu surface was ~48.1°, deviating from the typical value of 60° for Cu (111) (Fig. 1e, f)[15,25,26]. The cross-sectional annular dark-field scanning transmission electron microscopy image verifies the formation of the AS Cu surface with terrace and step-edge structures via the MS process (Fig. 1g). The Miller indices of the Cu films were random for each sample while preserving the SC film (Fig. S4), and a marginal variance was observed owing to the nature of the non-flat surface (Fig. S5). Similar SC AS metal (Cu or Au) films were obtained when the PC Mo foil replaced the PC W foil (Fig. S4).

### Orientation-analysis of SC AS metal films
We extended our examination of the lattice orientations of SC AS Cu films on PC W substrates by employing cross-sectional TEM. To examine both the Cu and W regions, a specimen from the edge of the Cu film on the PC W foil was prepared using a focused ion beam (FIB) (Inset of Fig. 2a). A low-magnification TEM image shows the AS Cu

layers on the PC W foils (Fig. 2a). The high-resolution TEM (HR-TEM) images obtained from regions I–III in Fig. 2a distinctly display the lattice fringes of the Cu(311) planes, as corroborated by the Joint Committee on Powder Diffraction Standards (JCPDS) reference data[27]. The fast Fourier transform (FFT) images corresponding to these regions showed a similar pattern (Fig. 2b). Furthermore, the selected area electron diffraction (SAED) patterns of the Cu film in regions IV–VI are mutually congruent (Fig. 2c). More importantly, these patterns have the same lattice orientation as those of the FFT patterns of regions I–III about the same zone axis as Cu[013]. This evidence substantiates that Cu layers maintain their SC structure throughout the measured area. Conversely, the four SAED patterns from regions VII–X in the PC W foil have various spot shapes and tilted angles with respect to each other (Fig. 2d). This demonstrates that the W foils retained a PC structure, which is in agreement with the EBSD results. A similar trend was observed for the Cu film on the Mo foils (Fig. S6).

### Theoretical models for AS metal films during MS process
To understand the formation mechanism of the SC AS metal film, two simple models for the solidification of metal overlayers on the PC W substrate were designed (Fig. 3a): (i) PC overlayer metal-film formation, which entails an epitaxial metal overlayer on each W grain but a grain boundary (GB) appears between two neighboring grains (Fig. 3a, (i) and (ii)) SC metal film formation, wherein the overlayer metal film is epitaxial to one W grain where the nucleation starts but no epitaxial relationship exists between the overlayer film and other W grains (Fig. 3a, (ii)). To simplify our analysis, we assume that the epitaxial and non-epitaxial interfaces between the metal overlayer and W substrate

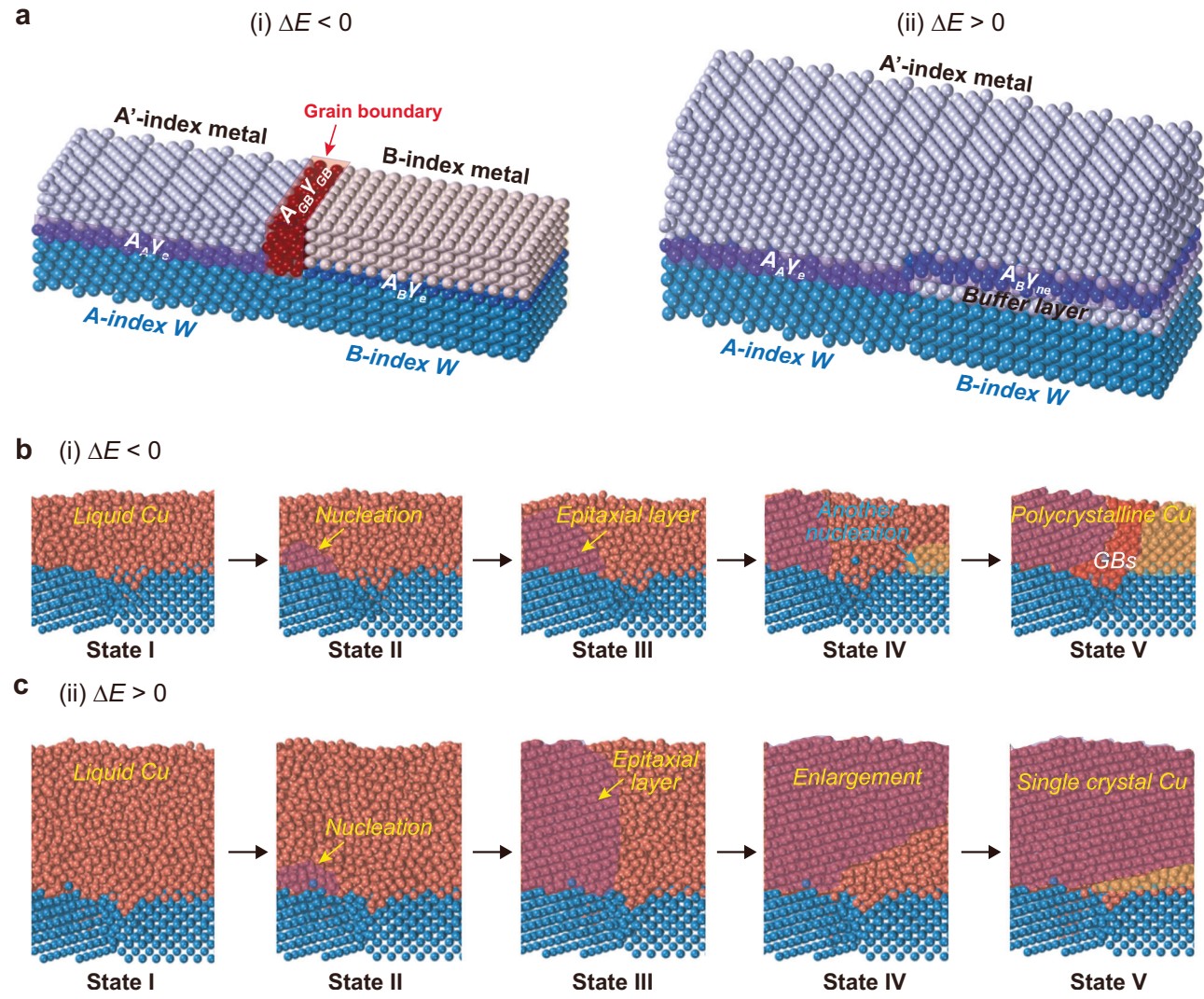

**Fig. 3 | Theoretical models for the formation of PC and SC AS metal films based on molecular dynamics (MD) simulations during the MS process. a** Schematic of two different models: (**i**) the formation of two different metal grains (A'-index metal and B-index metal) and (**ii**) development of SC metal (A'-index metal) film formed on a PC W surface (A-index and B-index). The total energy for each model is given by $E_1 = A_A\gamma_e + A_B\gamma_e + A_{GB}\gamma_{GB}$ and $E_2 = A_A\gamma_e + A_B\gamma_{ne}$. The energy difference between the models is $\Delta E = E_1 - E_2 = A_{GB}\gamma_{GB} - A_B(\gamma_{ne} - \gamma_e)$. A and $\gamma$ denote the area and interface energy of each index and grain boundary, respectively. Depiction of the solidification process from states I to V for (**b**), a thin Cu film (**i**, $\Delta E < 0$) and (**c**), a thick Cu film (**ii**, $\Delta E > 0$) on W(255) and W(001) surfaces.

have the average formation energies of $\gamma_e$ and $\gamma_{ne}$, respectively. We denote the average formation energy of the GBs in the overlayer PC metal as $\gamma_{GB}$. Thus, the total formation energies of the PC and SC overlayer metal films ($E_1$ and $E_2$) are

$$E_1 = A_A\gamma_e + A_B\gamma_e + A_{GB}\gamma_{GB} \quad (1)$$

$$E_2 = A_A\gamma_e + A_B\gamma_{ne} \quad (2)$$

where $A_i$ ($i = A, B$) is the surface area of each W grain and $A_{GB}$ is the interfacial area between the two overlayer metal grains. The formation energy difference between the SC and PC overlayer metal films is

$$\Delta E = E_1 - E_2 = A_{GB}\gamma_{GB} - A_B(\gamma_{ne} - \gamma_e) \quad (3)$$

Eq. (3) clearly shows that the formation of the overlayer film depends on the interface energies ($\gamma_{ne}, \gamma_e, \gamma_{GB}$), grain size of PC W ($A_i$), and thickness of the overlayer film (i.e., a thicker overlayer film leads to

a larger $A_{GB}$). From Eq. (3), when $\Delta E > 0$, we obtain

$$A_{GB}/A_B > (\gamma_{ne} - \gamma_e)/\gamma_{GB} \quad (4)$$

This implies that the SC overlayer film is energetically favorable if $A_{GB}/A_B$ is sufficiently large. $A_{GB}$ is proportional to the thickness of the overlayer film, and $A_B$ depends on the grain size of the PC W substrate. Thus, Eq. (4) clearly shows a competition between the overlayer thickness and PC W grain size, and the condition for forming the SC overlayer metal film is that the thickness of the overlayer metal ($h$) significantly larger than the grain size of the W substrate ($d$). To verify this conclusion, we built atomic models of Cu (or Au) overlayers with different thicknesses on PC W substrates and performed MD simulations to investigate the solidification of the overlayer films (Fig. 3b, c). The W(001) and (255) grains were used to build a polycrystalline W substrate. The effective thickness of the overlayer metals $h_{eff}$ is defined by $h_{eff} = \rho_S/\rho_V$, where $\rho_S$ and $\rho_V$ are the area and volume density of the overlayer metal atom, respectively. As clearly shown in our MD simulations, if the Cu overlayer is sufficiently thin (i.e., $h_{eff} = 1.5$ nm in our simulation), the epitaxial solidification of the Cu layer first nucleates on

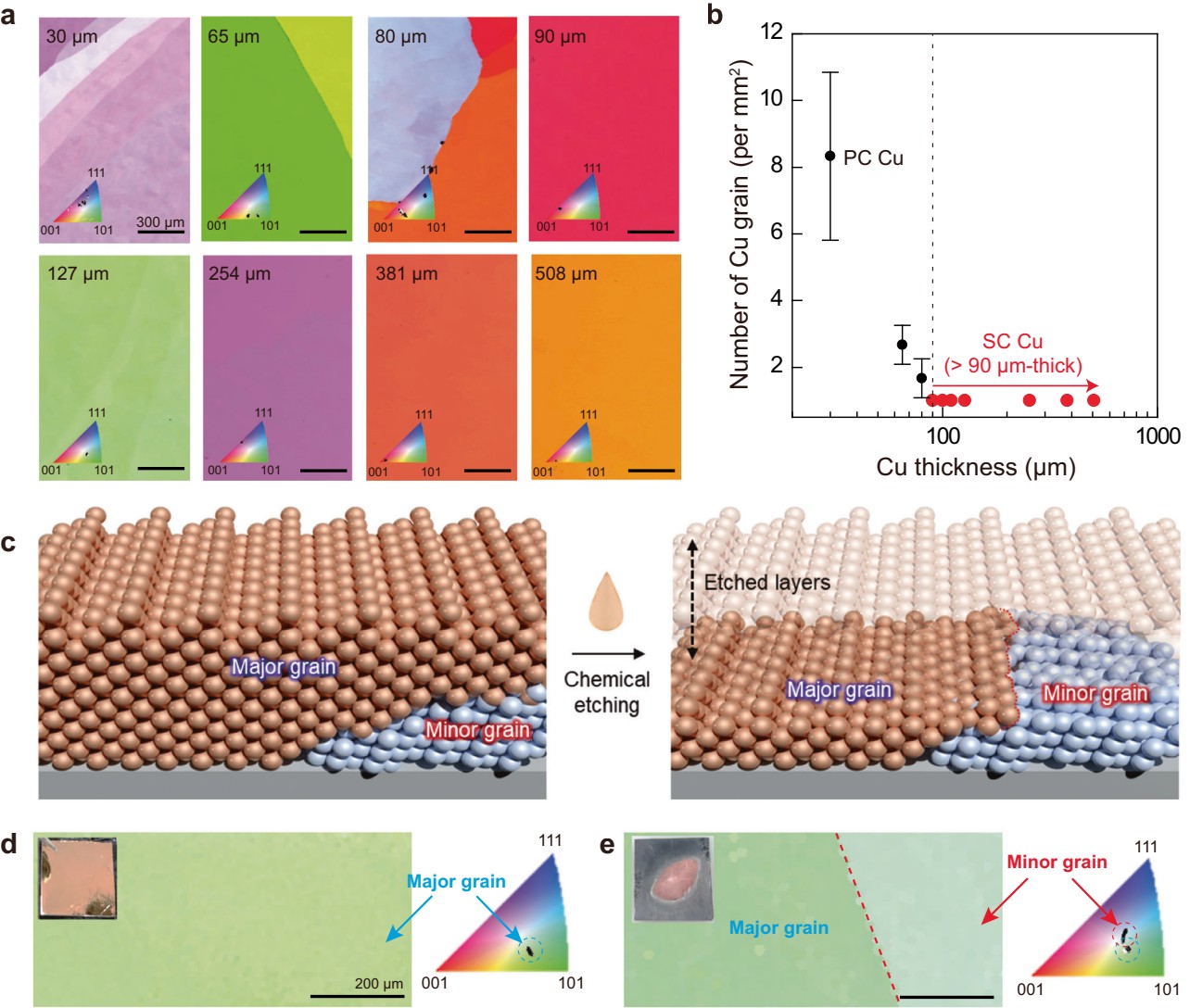

**Fig. 4 | Effect of Cu thickness on the formation of an SC AS Cu film. a** EBSD mapping images of the solidified Cu films with various thicknesses of pristine Cu foils. **b** The number of Cu grains on each sample in (**a**). The error bars in (**b**) represent the standard error. **c** Schematic of the chemical etching process to reveal the minor grain underneath the major grain in the SC AS Cu film. **d, e** EBSD mapping images with IPF maps of the SC AS Cu film before/after chemical etching.

the high-index W(255) surface (see states I–III in Fig. 3b); however, such an epitaxial SC layer cannot propagate onto the neighboring W(001) surface (Fig. 3b). Subsequently, the epitaxial nucleation and growth of a new Cu grain on the neighboring W (001) surface occurs, leading to a PC overlayer film (see states IV and V in Fig. 3b). In contrast, if the Cu film is sufficiently thick ($h_{eff}$ = 3.0 nm in our simulation), the epitaxial Cu film on the W(255) facet can easily propagate to the neighboring W(001) surface and eventually form the SC AS Cu film on the entire substrate (Fig. 3c). In this state, a disordered interface Cu layer, indicated as buffer layer, is formed between the single crystal Cu layer and W(001) surface. The formation of buffer layers, rather than secondary nucleation, may be attributed to the rapid propagation of the epitaxial Cu layer in thick Cu films, extending to neighboring W(001) surfaces, thereby disrupting secondary nucleation. This rapid propagation is also evident in the system energy changes (Fig. S7). Similar trends are also observed for different W facets (Figs. S8 and S9). Beyond the Cu overlayer, the solidification of the Au overlayer also exhibits a similar behavior (Fig. S10).

We further simulated the effect of the size of the PC W grains (i.e., $A_B$) on the solidification of the overlayer film, while maintaining the system size and thickness of the metal (Fig. S11). We set the effective

thicknesses as 3.00 and 4.17 nm for Cu and Au, respectively. These thicknesses were sufficient to form SC overlayer metal films. For comparison, the areal ratios of W(255) to (001) were set as A(255): A(001) = 2:8, 3:7, and 5:5. Both the molten Cu and Au layers become SC AS films with increasing A(255) area. Specifically, SC AS Cu layers are formed beyond A(255): A(001) = 3:7; however, Au layers require more than A(255): A(001) = 3:7 for the SC AS film. These results imply that the formation of SC AS Cu films (or films with high-index surfaces) is easier than that of Au films.

## Effect of Cu thickness for SC AS Cu film

To experimentally investigate the influence of metal thickness on the formation of SC AS metal films, the MS process was applied to Cu layers with thicknesses ranging from 3 to 508 μm. Cu layers up to 80 μm in thickness exhibited a multitude of Cu grains (Figs. 4a, b, and S12). Remarkably, Cu layers exceeding 90 μm in thickness displayed a single grain. This thickness dependence of the formation of SC Cu layers is in good agreement with theoretical predictions. To obtain a more detailed understanding, we chemically etched the SC Cu films using a Cu etchant to assess the crystal structure of Cu near the W substrate (Fig. 4c). The EBSD image obtained after partial etching of

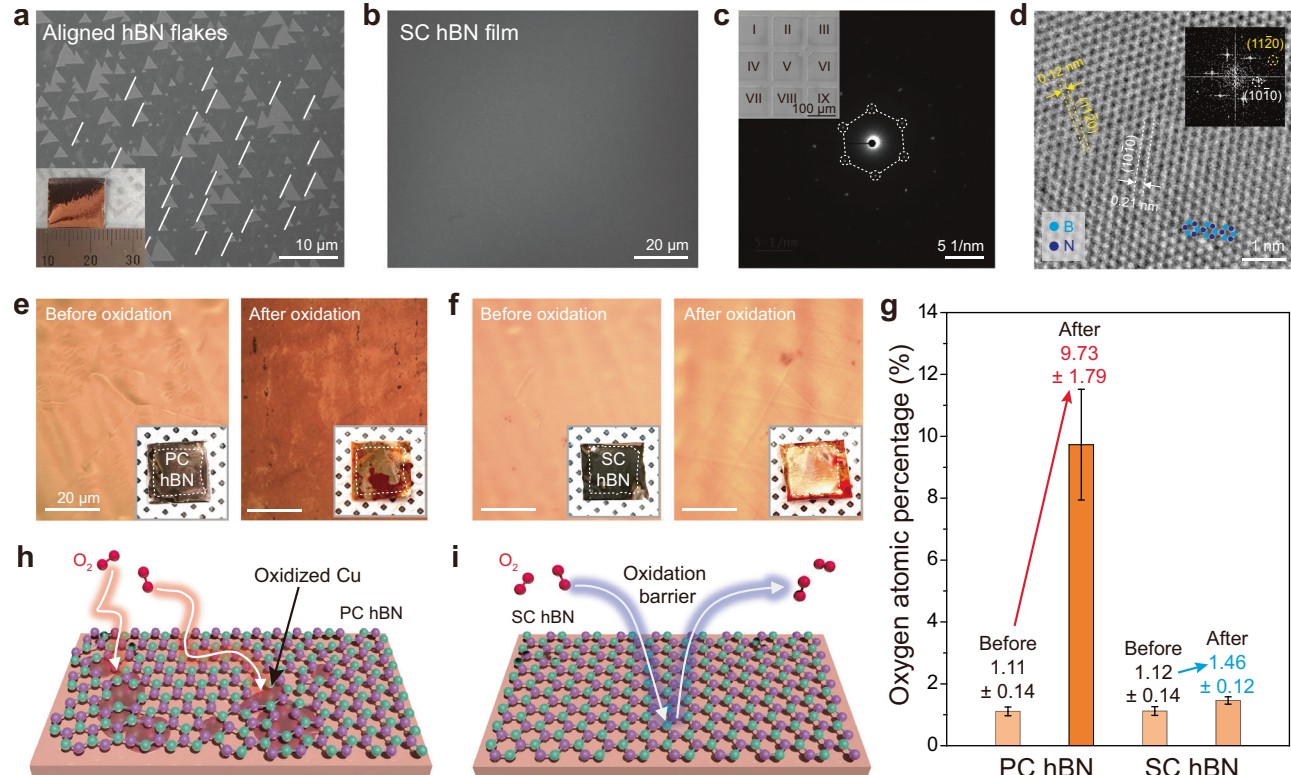

**Fig. 5 | Growth of SC hBN on SC AS Cu film and demonstration of the single-atom-thick oxidation barrier. a, b** SEM images of coherently aligned hBN grains and SC hBN film grown on SC AS Cu films. Inset in (**a**) shows a photograph of the centimeter-scaled sample. **c** Nine stacked SAED patterns obtained from nine different regions in 300 × 300 μm$^2$ (I–IX in inset). The inset in (**c**) shows a low-magnification TEM image of transferred SC hBN on a TEM grid. **d** An HR-TEM image of the SC hBN film (inset: corresponding FFT image). Cyan- and navy-colored balls indicate B and N atoms, respectively. **e, f** Optical images and photographs of transferred PC and SC hBN films on Cu foils before and after oxidation. **g** Atomic percentages of oxygen in the PC and SC hBN-covered Cu foils before and after oxidation. **h** A schematic of the oxidation behavior on PC and SC hBN-covered Cu foils.

the major Cu layer revealed two grains (i.e., major and minor grains). The corresponding IPFs confirmed the existence of the minor grain alongside the major grain, suggesting the presence of smaller Cu grains beneath the predominant Cu grain. This experimental observation aligns with our simulation results. While polycrystalline Cu forms near the polycrystalline W surface, the topmost Cu layers are observed to be single crystalline. This phenomenon can be attributed to a predominant Cu layer nucleating at a specific W facet and then entirely covering the Cu and W grains. Consequently, this process leads to the formation of SC Cu films (Fig. 3c).

## Growth of SC 2D materials on SC AS metal surfaces with oxidation barrier applications

It has been observed that 2D grains grown on the AS surface are coherently aligned with each other and these aligned 2D grains coalesce seamlessly, resulting in the formation of large-area SC 2D films with the presence of grain boundaries[11–23]. As a proof-of-concept for the SC growth platform of 2D vdW materials, SC AC Cu films were employed to grow SC hBN films. An SC hBN film was grown on a Cu (124) surface using borazine precursors at 1050 °C (Fig. S13). For 2 min 30 s of growth, more than 96% of hBN grains with an area greater than 1.3 × 1.3 cm$^2$ were coherently aligned (Figs. 5a and S14). SC hBN films were eventually synthesized with a prolonged growth for 4 min (Fig. 5b). The characteristic E$_{2g}$ phonon mode in the Raman spectrum clearly demonstrated the presence of hBN films (Fig. S13). The HR-TEM measurements were performed after transferring the monolayer hBN films onto a TEM grid. The hexagonal spots of nine SAED patterns of the hBN film, obtained from different regions over a 300 × 300 μm$^2$ area, completely overlap, indicating that these regions maintain single crystallinity

(Fig. 5c). Moreover, the HR-TEM image clearly shows lattice fringes of the (11-20) and (10-10) planes and a hexagonal structure with B and N atoms (Fig. 5d). Unlike PC hBN, which allows the permeation of gas molecules through its defective grain boundaries, SC hBN films have been demonstrated to be an ideal gas barrier[10,28–30]. Hence, the SC hBN film grown on AS Cu surface was further evaluated as oxidation barrier for Cu surfaces. Using a conventional polymethyl methacrylate (PMMA) method, both PC and SC monolayer hBN films were transferred onto Cu foils and subsequently oxidized at 250 °C in air for 30 min. Although the Cu surface underneath the PC hBN was severely damaged after oxidation, the surface protected by the SC hBN film remained almost unaffected (Fig. 5e, f). Energy-dispersive spectroscopy revealed that the atomic percentage of oxygen on the PC hBN-covered Cu surface increased approximately ninefold, whereas the change on the SC hBN-covered surface was negligible (Fig. 5g). This suggests that the SC hBN films serve as an effective barrier against the surface oxidation of the Cu foils. In contrast, the PC hBN films, with their inherent structural defects, including grain and twin boundaries, appeared to be more susceptible to oxidation (Fig. 5h). Finally, we replicated the synthesis of SC graphene and WS$_2$ on SC AS Cu and Au layers on PC W substrates (Fig. S15). Although different Miller indices were observed compared to previous studies[18,19], both graphene and WS$_2$ grains exhibited coherent alignment, ultimately leading to the growth of SC 2D materials. These findings strongly suggest that SC AS metal films can function as universal growth platforms for SC vdW 2D materials.

In summary, we reveal the mechanism of SC AS metal film formation on PC metal substrates during an MS process. The epitaxial solidification of the molten Cu layer nucleates in high-index W grains and easily extends to other W grains with lower interface energies. This

process result in a random Miller index of the SC AS Cu film on the PC W substrates. This randomness is attributed to the inherent variability of the W facts, where the Cu layer initially nucleated on the poly-crystalline W facets. Moreover, the thickness of the Cu overlayer is required to significantly exceed the average grain size of the W sub-strate. A similar trend is observed for Au on W (or Mo). As a proof-of-concept for the SC growth platform of 2D materials, we synthesize SC hBN films on the SC AS Cu surface, demonstrating their potential as oxidation barrier materials. The SC AS metal surface on a PC substrate, which is readily produced via the MS process, presents a potent plat-form for the large-scale growth of SC vdW 2D materials. This holds significant promise for the development of SC vdW heterostructures in the near future.

## Methods

### Cleaning procedures for metal foils
The as-received metal foils, including Cu (0.127 mm thick, 99.99%, Alfa Aesar), Au (0.2 mm thick, 99.99%, iNexus, Inc.), W (0.1 mm thick, 99.95%, Alfa Aesar), and Mo (0.1 mm thick, 99.95%, Alfa Aesar), were washed with acetone and isopropyl alcohol before being dried with $N_2$ gas. The precleaned Cu and Au foils were then cut into $1.2 \times 1.2$ cm$^2$ pieces, while W and Mo foils were cut into $1.3 \times 1.3$ cm$^2$ pieces. The Cu and Au foils underwent additional cleaning via mild sonication in Cu (type CE-100, Transene) and Au (GE-8111, Transene) etchants for 10 min. Finally, all metal foils were treated with a Ni etchant (TFB, Transene) for 5 min to eliminate residual metal particles[31]. Any etchant residue was rinsed with fresh deionized water.

### Preparation of SC AS Cu and Au films via the MS process
The cleaned Cu and Au foils were stacked on a W (or Mo) foil and placed at the center of a two-inch quartz tube. The tube was then purged of residual gases using high-purity Ar (>99.9999%) and $H_2$ (>99.999%) at flow rates of 1000 and 30 sccm for 5 min, respectively. Subsequently, the furnace temperature was increased to 1100 °C for 1 h and maintained at this temperature for an additional 5 min under the same Ar and $H_2$ atmosphere. The furnace was then cooled from 1100 °C to 1050 °C at a cooling rate of 0.18 °C/min, facilitating the formation of SC AS metal films.

### Growth of the SC monolayer hBN film
To grow the hBN film, a CVD system equipped with a bubbler con-taining a liquid borazine precursor was employed[32]. The SC AS Cu film on a PC W substrate (for the SC hBN film) or cleaned PC Cu foil (for the PC hBN film) was placed at the center of a two-inch quartz tube. The system was evacuated for 15 min using a dry pump. The furnace tem-perature was increased to 1050 °C for 40 min under an Ar and $H_2$ atmosphere at flow rates of 100 and 20 sccm, respectively. The tem-perature was maintained for 1 h before borazine was supplied at a flow rate of 0.01 sccm for 4 min to synthesize the hBN films. After the growth, the sample was naturally cooled to room temperature while maintaining the same flow rates of Ar and $H_2$.

### Growth of SC monolayer graphene
SC AS Cu films were employed for the synthesis of SC graphene. Upon loading, the growth substrate was purged with Ar gas at a flow rate of 100 sccm at atmospheric pressure. Subsequently, the temperature was increased to 1050 °C for over 40 min and the substrate was then annealed for 1 h with Ar and $H_2$ gases at flow rates of 100 and 5 sccm, respectively. Following the annealing process, $CH_4$ was introduced at a flow rate of 0.02 sccm for 20 min. The system was then allowed to cool naturally.

### Growth of SC monolayer WS$_2$
SC AS Au films were used for the growth of SC monolayer WS$_2$. First, 2 wt% of sodium tungstate dihydrate (Na$_2$WO$_4$•2H$_2$O, ≥99%, Sigma

Aldrich) in acetylacetone was spin-coated onto the growth substrate at 3000 rpm for 60 s. The coated substrate was then loaded into a quartz tube and subjected to a purge with high purity Ar (99.9999%) at a flow rate of 1000 sccm for 5 min. Thereafter, the furnace temperature was increased to 850 °C and sustained with a supply of (NH$_4$)2S at a flow rate of 2 sccm for 10 min. Upon completion of the growth, the furnace was allowed to cool naturally to room temperature. The entire pro-cedure was performed in Ar and $H_2$ atmospheres at flow rates of 400 and 10 sccm, respectively.

### Transfer of SC hBN films grown on SC AS Cu/PC W substrates
Initially, poly (methyl methacrylate) (A9 PMMA, MicroChem) was applied to the as-grown SC hBN films via spin-coating at 3000 rpm for 1 min. The coated films were baked in an oven at 90 °C for 5 min. During detachment, the PMMA-coated SC hBN films were connected to a cathode, whereas the Pt foil was linked to an anode. A voltage of 3 V was applied to a 1 M NaOH aqueous solution to release the PMMA/ hBN films from the growth substrate. Once detached, the films were thoroughly rinsed with deionized water before being transferred to the target substrates (e.g., TEM grids or Cu foils). Finally, the PMMA layer was removed by dipping the substrate in acetone for 30 min.

### Oxidation treatment of transferred PC and SC hBN films on Cu foils
The transferred PC and SC hBN films on the Cu foils were loaded in a CVD chamber. The chamber was heated to 250 °C and maintained at this temperature for 30 min in an air atmosphere. After oxidation, the chamber was opened and cooled to room temperature.

### Characterization
The surface morphologies of the samples were analyzed using field-emission scanning electron microscopy (FE-SEM; JSM7600F, JEOL, Japan). The crystal orientations of the metals were characterized via EBSD using FE-SEM (JSM7000F, JEOL). The phonon mode of hBN was measured using a Raman spectroscopy system equipped with a 532 nm laser (NTEGRA Spectra, NT-MDT). The crystal structures and lattice orientations of the hBN films were evaluated using scanning trans-mission electron microscopy (STEM; JEM ARM 200F, JEOL, Japan) and polarized optical microscopy (Eclipse LV150, Nikon). The lamellae of the Cu/W and Cu/Mo samples for the cross-sectional STEM measure-ments were prepared using a FIB (FEI Helios NanoLab 450, Eurofins Nanolab Technologies).

### System setup for the MD simulations
To understand the formation mechanism of the SC metal film on the PC W substrate, MD simulations were conducted using the large-scale atomic/molecular massively parallel simulator (LAMMPS) software package[33]. In the MD simulations, the embedded atom method was employed to characterize interactions between same metal atoms (such as Cu-Cu, Au-Au, and W-W)[34,35], while the Morse potential was utilized to represent interactions between different metal atoms (Cu-W and Au-W)[36]. The Morse potential is defined as follows:

$$E_{Morse} = \sum_{i \neq j} D_e \left[ e^{-2\alpha(r_{ij}-r_e)} - 2e^{-\alpha(r_{ij}-r_e)} \right] \tag{5}$$

where $r_{ij}$ is the interatomic distance between $i$ and $j$ atoms, $D_e$ is the potential well depth, $r_e$ is the equilibrium distance, and $\alpha$ is a coeffi-cient. To parameterize the Morse potential, we calculated the inter-action between the metal layer and substrate with a distance range of 1.5–10 Å using the DFT method. The Perdew–Burke–Ernzerhof exchange-correlation functional and generalized gradient approxima-tion implemented in the Vienna ab initio Simulation Package were used[37–39]. The plane-wave cutoff was set to 600 eV, and the convergence criterion for the self-consistent field iterations was set to 10$^{-5}$ eV. The optimizations of the lattice cell and atomic positions

were performed until the convergence of the forces on all atoms was less than 0.01 eV/Å. Vacuum slabs with more than 15 Å were used, and the Brillouin zone was sampled with a k-points density of 0.03 Å$^{-1}$. In this study, the fitting parameters for Cu/Au on the W substrate were $D_e = 0.37/0.42$ eV, $r_e = 2.79/3.02$ Å, and $\alpha = 1.50/1.65$ Å$^{-1}$. The in-plane supercell size for all MD simulations was 100 Å × 30 Å, and the thickness of the supercell depended on the simulations. The thickness of the PC W substrate was set at 14 Å, and W atoms of the bottom layer in the substrate were fixed during simulations. We performed MD simulations with an NVT ensemble using a Nose–Hoover style thermostat[40–42]. The system temperature was set to quickly increase from 300 to 1200 K in 1 ns and then quickly decrease to 1000 K in 1 ns. The temperature then slowly decreased to 800 K (near freezing temperature) in 10 ns, which was the annealing process, and finally, the temperature quickly decreased to 300 K in 1 ns.

## Data availability

The authors declare that all data supporting the findings of this study are available within the paper and its supplementary information files. Source data are provided with this paper.

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

## Acknowledgements

K.K.K. acknowledges support from the Basic Science Research Program and Next-Generation Intelligence Semiconductor Program through the National Research Foundation of Korea (NRF), funded by the Ministry of Science, ICT & Future Planning (2022R1A2C2091475, 2022M3F3A2A01072215). S.M.K. acknowledges support from the Basic

Science Research Program through the NRF, funded by the Ministry of Science, ICT & Future Planning (2022R1A2C2009292, 2022M3H4A1A04096396, 2022R1A4A3030766). S.H.C., K.K.K., and Y.H.L. acknowledge support from the Institute for Basic Science (IBS-R011-D1). F. D acknowledges the support of the National Science Foundation of China (NSFC-22333005), High-Talent Support from the Shenzhen Institute of Advanced Technology (SIAT-SE3G0991010, 2023), and a Startup Research Grant from the Shenzhen Institute of Advanced Technology. Y. P. and S. Y. K. acknowledge support from the Basic Research Laboratory (BRL) program (2021R1A4A1033224) of the NRF and thank the UNIST Supercomputing Center for providing access to its supercomputing resources. This study was supported by the Advanced Facility Center for Quantum Technology.

## Author contributions

H.K., S.H.C., and Y.P. contributed equally to this work. H.K. and S.H.C. designed and experimented. Y.P. performed the MD simulation. S.J.L., C.S.O., S.Y.K. supported the experimental parts. Y.H.L., S.M.K., F.D., and K.K.K. designed and the developed this work. All first co-authors and corresponding authors participated in the writing of manuscripts. H.K., S.H.C., Y.P., S.M.K., Y.H.L., F.D., and K.K.K. drew the figures and wrote the manuscript. All authors participated in the manuscript review.

## Competing interests

The authors declare no competing interests.
