## [Transparent Peer Review file · Nature Communications]

Atomic sawtooth-like metal films for vdW-layered single-crystal growthREVIEWER COMMENTS

Reviewer #1 (Remarks to the Author):

The authors present a methodology for the production of large-scale single-crystal Cu/Au films with a high-index ("sawtooth") surface on top of the polycrystalline W/Mo films. Further, they propose the mechanism of the single-crystal formation as shown with MD simulations, and show the effectiveness of the method by using the produced film as a substrate for the growth of the monocrystal hBN.

The use of the vicinal substrate steps to align nuclei of 2D materials was recently shown to be effective, motivating the presented work, where the methodology appears effective and produces good results. However, authors need to elaborate on the importance of their contribution, and the theoretical modeling part has several aspects that must be clarified, corrected, or perhaps deleted not to blur the story. Based on that, I recommend a major revision.

- The term "sawtooth surface" is intuitive but uncommon, not defined in the manuscript and is used only once outside of the abstract and title. Vicinal surface is well established in science and technology; No need to introduce a novel term in this context.
- As wafer-scale growth of hBN was previously achieved on both vicinal and (111) surfaces (Ref.11-12), and the single crystalline Cu films were also achieved (Ref. 14), authors should elaborate on the importance of their contribution. I note a relevant paper which precedes all Refs 11-14 and must be cited, K Bets et al. Nano Lett.19, 2027 (2019).
- The proposed theoretical model has several significant issues:
 - Why, within the MD simulation, the secondary nucleation does not appear on the W(001) grain within the thicker Cu film? This process should not in any way be affected by the thickness of the film. The detection of the nuclei within the MD simulation is questionable, as a very regular lattice formed by Cu atoms on W(001) grain is visible in every single frame shown, but not highlighted as a nucleus.
 - It is extremely unlikely that the nucleation occurs on the single grain of W within a centimeter-scale polycrystalline film, which is confirmed by the experimental observation of minor grains. This is in no way addressed in the proposed mechanism.
 - The analytics (1-3) represents a thermodynamic equilibrium, not to be achieved within a 13 ns MD simulation. Furthermore, to address the concerns listed above, the $dE > 0$ case should include the secondary grain, GB region, etc.
 - In the method section, the sentence "In the MD simulations, the ... (EAM) potentials were used to describe the interactions between the overlayer metal atoms (Cu or Au) and W atoms, and the Morse potential was used to describe the interaction between the overlayer metal atoms and W atoms" fails to specify what Morse and EAM potentials were actually used for.
- The "...in which the symmetry group belongs to a subgroup of 2D vdW materials" motivation of the substrate surface choice is not convincing, unnecessary, and simply does not belong here, simpler is to remove this block lines 52-55. Following this bizarre logic, the best and most universal substrate would possess no symmetry at all and will work for the growth of all and any 2D material. Furthermore, for most metals at any reasonable growth temperature, the surface is unlikely to maintain the symmetry defined for the idealized structure. So this controversial idea is better to leave out.

Reviewer #2 (Remarks to the Author):

This paper reports the mechanism of the formation of Cu and Au single crystal films on polycrystalline W foil. The synthesis of single-crystalline catalyst metals is very important for the growth of high-quality 2D materials, such as hBN, TMD, and graphene.

In this manuscript the authors discuss the mechanism of the formation by combining the experimental work (metal thickness dependence (Fig. 4a,b) and the metal etching (Fig. 4c,d)) and theoretical analysis (Fig. 3). I think that the model proposed in Fig 3. is reasonable and well explains the single crystal metal formation by epitaxial solidification.

However, in terms of the aligned growth of 2D materials, demonstrated in Figs. 5 (hBN) and Fig.

S12 (graphene and WS₂), the presence of atomic sawtooth (AS) surface structure is essential, as demonstrated in the authors' previous paper (refs. 17,18). However, the present manuscript discusses mainly the formation of the single-crystalline metal films on a polycrystalline W foil. Therefore, the reviewer is not well convinced with the novelty and importance of this work. At least, the authors should discuss the mechanism of formation of atomic sawtooth (AS) surface structure in the single-crystal metal surface. Difference between the well-known step-terrace structure and the proposed AS structure is also unclear for the reviewer.

Overall, the reviewer finds it a borderline case for publication in Nature Communications with due to the high scientific standards of the journal. At least it needs more experimental and theoretical explanations on the mechanism of AS structure formation, which is an essential point of the high-quality 2D materials growth.

Reviewer #3 (Remarks to the Author):

This paper presents a combined experimental and theoretical study on the growth of single- and polycrystalline metals, growing 2D materials on top of the metals, and the oxidation barriers endowed by the grown h-BN on Cu. Energy analysis arguments and molecular dynamics simulations explained the experimental findings. The work demonstrates a novel approach to control the texture of metal layers on top of a metal substrate, and the theoretical insights can be extended to different material systems. Considering the potential impacts the current work will lead to, I recommend it for publication after the authors address my following concerns.

1. The logic links between the three parts of the contents should be strengthened. For example, the effects of AS structures, and the grain boundaries on the nucleation and growth of 2D materials, although maybe reported in previous studies, should be included for completeness. What controls the atomic-level structures of the AS Cu surfaces? The essence of single crystallinity in h-BN on the oxidation-barrier function lacks explanation from, for example, atomistic simulations.
2. The thickness dependence of single crystallinity is very interesting. Can the critical value of 90 micrometers be quantitatively estimated from theory since the molecular simulations are limited to nanometer thicknesses? Why the topmost layers are single crystalline (line 188)?
3. Crystal growth (of the metal layers) is a kinetic process. The molecular dynamics simulations were performed at the scale of nanometers, which may be sufficient for nanometer-thick layers but for sure are not enough for the micrometer ones. The authors should change the experimental processing conditions such as the temperature profiles to check the kinetic effects beyond the thermodynamic arguments in Eqs. 1-4.
4. There is a lot of information missing from the simulations. For example, why W(255) and W(001) surfaces were chosen? Why can the oxidation layers of Cu and W be neglected?

Reviewer #4 (Remarks to the Author):

This manuscript elucidates the principles behind the formation of atomic sawtooth metal film for the growth of single-crystal van der Waals (vdW) layered materials. Molecular dynamics simulations and experimental results were employed to investigate the formation mechanism of atomic sawtooth surface during the solidification process of molten metal on polycrystalline metal substrates. The findings underscore the importance of having a sufficient thickness of molten metal for the formation of atomic sawtooth film. Additionally, the authors demonstrated the growth and application of single-crystal vdW layered materials. Given that single-crystal growth of vdW layered materials is a crucial topic in the community, this manuscript introduces novelty suitable for publication in Nature Communications. The manuscript is logically well-organized, and the authors used appropriate research strategies to understand the mechanism. Therefore, I recommend publishing this manuscript in Nature Communications after minor revisions. I have provided some comments below to enhance the completeness of this work.

1. The authors mentioned that "The epitaxial solidification of the molten Cu layer nucleated in

high-index W grains and easily extended to other W grains with lower interface energies. This process resulted in a random Miller index of the SC AS Cu film on the PC W substrates owing to the inherent randomness of the W facets." This seems to be a bit inconsistent. Since the nucleation starts at a high-index surface and extends to form a single crystal, the Cu surfaces cannot be completely random; they should also prefer high-indices. Also, I wonder if there exists a more quantitative relation between the surface indices of W and Cu? e.g. if the highest index surface of the PC W is (211), would it result in (211) surface for SC Cu as well? What is the Cu surface index observed in the MD simulation?

2. The author performed hBN growth at temperatures exceeding 1000 oC. I am curious whether the index of the atomic sawtooth Cu film remains unchanged at high temperatures.

3. Would the cooling rate during the solidification process be an important factor?

Reviewer #1 (Remarks to the Author):

The authors present a methodology for the production of large-scale single-crystal Cu/Au films with a high-index (“sawtooth”) surface on top of the polycrystalline W/Mo films. Further, they propose the mechanism of the single-crystal formation as shown with MD simulations, and show the effectiveness of the method by using the produced film as a substrate for the growth of the monocrystal hBN.

The use of the vicinal substrate steps to align nuclei of 2D materials was recently shown to be effective, motivating the presented work, where the methodology appears effective and produces good results. However, authors need to elaborate on the importance of their contribution, and the theoretical modeling part has several aspects that must be clarified, corrected, or perhaps deleted not to blur the story. Based on that, I recommend a major revision.

Response: We would like to thank you for your thoughtful and valuable comments to improve our manuscript. We have thoroughly revised the manuscript based on reviewer’s comments by the clarification, correction, and deletion of the suggested parts. We believe that our responses would address all your concerns and that you will find our manuscript suitable for publication.

1. The term “sawtooth surface” is intuitive but uncommon, not defined in the manuscript and is used only once outside of the abstract and title. Vicinal surface is well established in science and technology; No need to introduce a novel term in this context.

Response: We appreciate the reviewer's understanding of the surface structures. As known, a vicinal surface comprises steps and terraces, typically created by mis-cutting a high-symmetry surface at a slight angle, resulting in a 'stair-like' appearance, as illustrated in Fig. R1. In contrast, while an atomic sawtooth structure also includes 'steps and terraces', its appearance is distinctly different, resembling an 'atomic sawtooth' rather than stairs, We hope the reviewer acknowledges this difference, which justifies our use of the term 'atomic sawtooth surface' as described in our previous paper [S. H. Choi et al. Adv. Mater. 33, 2006601 (2021)]. We kindly ask for your understanding in this matter.

Figure R1. Schematics of (a) stair-like vicinal surface and (b) atomic sawtooth surface.

2. As wafer-scale growth of hBN was previously achieved on both vicinal and (111) surfaces (Ref.11-12), and the single crystalline Cu films were also achieved (Ref. 14), authors should elaborate on the importance of their contribution. I note a relevant paper which precedes all Refs 11-14 and must be

cited, K Bets et al. Nano Lett.19, 2027 (2019).

Response: We acknowledge the importance of the references mentioned, which we discussed from page 3, lines 23-33. We would appreciate it if the reviewer could kindly review this section again for clarification. Additionally, we are grateful for the reviewer's suggestion of a relevant paper that was previously not included in our manuscript. In response, we have now added the reference [K. Bets et al., Nano Lett., 19, 2027 (2019)] and incorporated additional sentences in the revised manuscript to reflect this.

- In line 20 of page 3: “The importance of vicinal surfaces for SC growth has also been proposed through studies using density functional theory and molecular dynamics¹¹.”

3. The proposed theoretical model has several significant issues: Why, within the MD simulation, the secondary nucleation does not appear on the W(001) grain within the thicker Cu film? This process should not in any way be affected by the thickness of the film. The detection of the nuclei within the MD simulation is questionable, as a very regular lattice formed by Cu atoms on W(001) grain is visible in every single frame shown, but not highlighted as a nucleus.

Response: We appreciate the valuable comments. We also agree that secondary nucleation occurs irrespective of the thickness of Cu film. However, it was not observed in our MD simulation results for the thick Cu film. This maybe attribute to differences in nucleation times between thin and thick Cu films. Upon examining the system energy changes (as shown in Figures R2a and b), we observed that the first Cu nucleation occurs at a similar time (state II). However, in the thick Cu film, the solidification progresses more rapidly to the neighboring W(001) facet (states III and IV), thereby disrupting secondary nucleation (as shown in Figures R2c and d). Consequently, even at state V, a disordered layer was observed. Given the difficulty in describing this disordered layer as secondary nucleation, we have revised the manuscript to classify it as a buffer layer instead of non-epitaxial layer and have provided an explanation for this issue.

Figure R2 | System energy changes and the simulated results of the MD simulations during the MS process. a,c, thin and b,d, thick Cu films on a PC W surface.

- In line 2 of page 7: “In this state, a disordered interface Cu layer, indicated as buffer layer, is formed between the single crystal Cu layer and W(001) surface. The formation of buffer layers, rather than secondary nucleation, may be attributed to the rapid propagation of the epitaxial Cu layer in thick Cu films, extending to neighboring W(001) surfaces, thereby disrupting secondary nucleation. This rapid propagation is also evident in the system energy changes (Fig. S7).”

Figure S7 | System energy changes of the MD simulations during the MS process for **a**, thin and **b**, thick Cu films on a PC W surface. The states indicate each phase change in Figures 3b and c.

Figure 3 | Theoretical models for the formation of PC and SC AS metal films based on molecular dynamics (MD) simulations during the MS process. **a**, Schematic of two different models: (i) the formation of two different metal grains (A'-index metal and B-index metal) and (ii) development of SC metal (A'-index metal) film formed on a PC W surface (A-index and B-index). The total energy for each model is given by $E_1 = A_A\gamma_e + A_B\gamma_e + A_{GB}\gamma_{GB}$ and $E_2 = A_A\gamma_e + A_B\gamma_{ne}$. The energy difference between the models is $\Delta E = E_1 - E_2 = A_{GB}\gamma_{GB} - A_B(\gamma_{ne} - \gamma_e)$. A and γ denote the area and interface energy of each index and grain boundary, respectively. **b,c**, Depiction of the solidification process from states I to V for **b**, a thin Cu film (i, $\Delta E < 0$) and **c**, a thick Cu film (ii, ΔE

> 0) on W(255) and W(001) surfaces.

4. It is extremely unlikely that the nucleation occurs on the single grain of W within a centimeter-scale polycrystalline film, which is confirmed by the experimental observation of minor grains. This is in no way addressed in the proposed mechanism.

Response: Our experimental data clearly demonstrate the formation of single crystalline Au/Cu metal films on polycrystalline W films. While this result might initially seem extremely unlikely, our works provide robust evidence to the contrary. To elucidate this phenomenon, we developed a simplified model focusing on the formation of metal layers on two distinct types of W grains.

In our molecular dynamics (MD) simulations, it was observed that epitaxial Cu/Au layers predominantly nucleate on high-index W grains. This preference is attributed to the higher interfacial energy associated with these grains. Following nucleation, the Cu/Au layers propagate onto other W grains, ultimately leading to the formation of a single crystalline Cu/Au film over the entire surface, as depicted in Fig. 3c. While nucleation on other W grains is possible, these events are typically overshadowed by the more dominant Cu/Au layer that solidifies earlier on the high-index W grains. This model aligns well with our experimental observations, where the topmost metal layer is single crystalline, whereas the underlying layers are polycrystalline. This concurrence between our model predictions and experimental data underscores the validity of our approach.

We acknowledge the limitation that simulating a system on a centimeter scale is currently unfeasible, primarily due to computational constraints. However, we strongly believe that the scenario observed in our smaller-scale simulations is likely reflective of what occurs in larger systems. Our confidence in this extrapolation is based on the fundamental principles and behaviors observed in our model.

5. The analytics (1-3) represents a thermodynamic equilibrium, not to be achieved within a 13 ns MD simulation. Furthermore, to address the concerns listed above, the $dE > 0$ case should include the secondary grain, GB region, etc.

Response: We agree with the reviewer that MD simulation of 13 ns is significantly shorter than the time scale of real experiments, which is from minutes to hours. However, the sizes of our simulated systems, which contain a few thousand atoms, are also significantly smaller than that of the real experimental materials. Based our experience, such a smaller system can reach thermal equilibrium in a much shorter time scale, such as 100 ps. So, we believe that our simulation time of 13 ns is sufficient for this study. In addition, we have added system energy changes in MD simulation as Fig. S7 and the following explanations in the revised manuscript for secondary nucleation for the $dE > 0$

case.

- In line 2 of page 7: “In this state, a disordered interface Cu layer, indicated as buffer layer, is formed between the single crystal Cu layer and W(001) surface. The formation of buffer layers, rather than secondary nucleation, may be attributed to the rapid propagation of the epitaxial Cu layer in thick Cu films, extending to neighboring W(001) surfaces, thereby disrupting secondary nucleation. This rapid propagation is also evident in the system energy changes (Fig. S7).”

Figure S7 | System energy changes of the MD simulations during the MS process for a, thin and b, thick Cu films on a PC W surface. The states indicate each phase change in Figures 3b and c.

6. In the method section, the sentence “In the MD simulations, the ... (EAM) potentials were used to describe the interactions between the overlayer metal atoms (Cu or Au) and W atoms, and the Morse potential was used to describe the interaction between the overlayer metal atoms and W atoms” fails to specify what Morse and EAM potentials were actually used for.

Response: We appreciate the valuable comments. The EAM potentials were utilized to describe interactions between same metal atoms (such as Cu-Cu, W-W etc.), whereas the Morse potential was employed to represent interactions between different metal atoms (Cu-W). To avoid misleading, we have included additional sentences in the revised manuscript.

- In line 1 of page 12: “In the MD simulations, the embedded atom method was employed to characterize interactions between same metal atoms (such as Cu-Cu, Au-Au, and W-W)^{34,35}, while the Morse potential was utilized to represent interactions between different metal atoms (Cu-W and Au-W)³⁶.”

7. The “...in which the symmetry group belongs to a subgroup of 2D vdW materials” motivation of the substrate surface choice is not convincing, unnecessary, and simply does not belong here, simpler is to remove this block lines 52-55. Following this bizarre logic, the best and most universal substrate would possess no symmetry at all and will work for the growth of all and any 2D material. Furthermore, for most metals at any reasonable growth temperature, the surface is unlikely to maintain the symmetry defined for the idealized structure. So this controversial idea is better to leave out.

Response: As the reviewers are aware, the symmetry relationship between 2D materials and their growth substrate is significantly important. Moreover, it is essential that the growth substrate belongs to a subgroup of symmetry group of the 2D materials [Ref. 23]. This relationship is supported by numerous experimental observations [Refs. 11-19]. But we concur with the reviewer’s comment that the lowest-symmetry group is likely to be represented by the vicinal surface. Consequently, we have retained the sentences in our manuscript that provide guidance for designing the growth substrates and revised the last sentence to more clearly reflect this concept.

- In line 18 of page 3: “According to this concept, the lowest-symmetry substrate, featuring a vicinal surface (C_{1v}) is an ideal growth platform for SC 2D vdW materials²³.”

Overall, we appreciate the thoughtful comments and valuable suggestions from the reviewer. We have made a necessary revision to fully address the concerns raised by the reviewers, which greatly helped us to improve the manuscript. We believe that our study represents an important advancement in the related field thus it could make a valuable contribution to *Nature Communications*.

Reviewer #2 (Remarks to the Author):

This paper reports the mechanism of the formation of Cu and Au single crystal films on polycrystalline W foil. The synthesis of single-crystalline catalyst metals is very important for the growth of high-quality 2D materials, such as hBN, TMD, and graphene. In this manuscript the authors discuss the mechanism of the formation by combining the experimental work (metal thickness dependence (Fig. 4a,b) and the metal etching (Fig. 4c,d)) and theoretical analysis (Fig. 3). I think that the model proposed in Fig 3. is reasonable and well explains the single crystal metal formation by epitaxial solidification.

Response: We are grateful to the reviewer for thoroughly reviewing our manuscript and providing valuable comments. In response, we have prepared detailed point-by-point responses and revised the manuscript to address each of your insightful questions and suggestions. These revisions have significantly enhanced the quality of our manuscript.

1. However, in terms of the aligned growth of 2D materials, demonstrated in Figs. 5 (hBN) and Fig. S12 (graphene and WS₂), the presence of atomic sawtooth (AS) surface structure is essential, as demonstrated in the authors' previous paper (refs. 17,18). However, the present manuscript discusses mainly the formation of the single-crystalline metal films on a polycrystalline W foil. Therefore, the reviewer is not well convinced with the novelty and importance of this work.

Response: Previous reports [18,19] have established the synthesis mechanisms of single-crystal 2D vdW materials on atomic sawtooth (AS) surfaces. However, the formation process of these AS surfaces themselves have largely unexplored. Our work is specifically focused on revealing the underlying mechanism responsible for the formation of a single-crystal, atomic sawtooth (AS) metal films on polycrystalline W films during the melting-solidification (MS) process. Furthermore, we have extended this research to apply the AS Cu film, produced via the MS process, in the growth of single-crystal monolayer hBN film, which has not been previously reported. Additionally, our study demonstrates the use of single-crystal hBN film as an effective oxidation barrier, further highlighting the practical implications of our finding. We believe that the insights gained from this research provide a foundation groundwork for future studies aimed at controlling and exploiting atomic sawtooth surfaces. We hope that this explanation adequately clarifies the novelty and importance of our work, addressing the concerns of the reviewer.

2. At least, the authors should discuss the mechanism of formation of atomic sawtooth (AS) surface structure in the single-crystal metal surface.

Response: We appreciate the valuable comments for our study. The solidification process of molten metal undergoes stages analogous to the growth of materials including nucleation and growth. Therefore, the atomic-level structure of the AS Cu surface is mainly determined by the facets of W where the initial nuclei are formed during the solidification process. Therefore, we modified a sentence in the revised manuscript to address this point.

- In line 29 of page 7: “This experimental observation aligns with our simulation results. While polycrystalline Cu forms near the polycrystalline W surface, the topmost Cu layers are observed to be single crystalline. This phenomenon can be attributed to a predominant Cu layer nucleating at a specific W facet and then entirely covering the Cu and W grains. Consequently, this process leads to the formation of SC Cu films as illustrated in Fig. 3c.”

3. Difference between the well-known step-terrace structure and the proposed AS structure is also unclear for the reviewer.

Response: We appreciate the reviewer's understanding of the surface structures. As known, a vicinal surface comprises steps and terraces, typically created by mis-cutting a high-symmetry surface at a slight angle, resulting in a 'stair-like' appearance. However, while an atomic sawtooth structure also includes 'steps and terraces,' its appearance is distinctly different, resembling an 'atomic sawtooth' rather than stairs, as illustrated in Fig. R1. We hope the reviewer acknowledges this difference, which justifies our use of the term 'atomic sawtooth surface' as described in our previous paper [S. H. Choi et al. Adv. Mater. 33, 2006601 (2021)]. We kindly ask for your understanding in this matter.

Figure R1 | Schematics of (a) stair-like vicinal surface and (b) atomic sawtooth surface.

4. Overall, the reviewer finds it a borderline case for publication in Nature Communications with due to the high scientific standards of the journal. At least it needs more experimental and theoretical explanations on the mechanism of AS structure formation, which is an essential point of the high-quality 2D materials growth.

Response: We appreciate the reviewer's valuable interest on our study. Following reviewer's suggestion, we additionally performed the melting-solidification (MS) experiments with faster cooling

rates to offer the information of kinetic process of crystal growth as the experimental results for the mechanism of AS structure formation. Moreover, we also added the simulation results with three different W facets to support our mechanism and to support our studies as theoretical results. Thus, these have included the following experimental and theoretical results:

1. **Experimental results:** It is observed that as the cooling rate increased during the solidification process of molten Cu, more Cu grains were nucleated (as shown in Fig. S3 below). This aligns with the MD simulation results, which illustrate the formation of a single Cu grain initiating solidification, subsequently covering other areas before another nucleation occurs, thus resulting in a SC AS Cu film. Therefore, we added this result as Fig. S3 and discussed it in the revised manuscript.

- In line 2 of page 5: “It is noted that while the optimal cooling rate (< 0.2 °C/min) resulted in the formation of a SC AS metal film, faster cooling rates (3.3 and 16 °C/sec) led to the formation of PC AS metal film (Fig. S3). This observation may be attributed to the initiation of several nuclei simultaneously at multiple points during the rapid cooling process.”

Figure S3 | EBSD images with IPF maps of the solidified Cu surfaces with different cooling rates of **a**, fast cooling (-3.3 °C/s) and **b**, quenching cooling (-16 °C/s).

2. **Simulation results:** Even when three W facets are present, epitaxial Cu film formation was initially observed on the facet where nucleation occurred. This finding further validates the reliability of our simulation results previously obtained with two W facets. Therefore, we also added this result as Fig. S8 to support our proposed mechanism.

- In line 7 of page 7: “Similar trends are also observed for different W facets (Figs. S8 and S9).”

Figure S9 | MD simulation results of Cu/W system with different three W indices. Simulated results about different three W indices with **a, thin Cu and **b**, thick Cu.**

Overall, we appreciate the thoughtful comments and valuable suggestions from the reviewer. We have made a necessary revision to fully address the concerns raised by the reviewers, which greatly helped us to improve the manuscript. We believe that our study represents an important advancement in the related field thus it could make a valuable contribution to *Nature Communications*.

Reviewer #3 (Remarks to the Author):

This paper presents a combined experimental and theoretical study on the growth of single- and polycrystalline metals, growing 2D materials on top of the metals, and the oxidation barriers endowed by the grown h-BN on Cu. Energy analysis arguments and molecular dynamics simulations explained the experimental findings. The work demonstrates a novel approach to control the texture of metal layers on top of a metal substrate, and the theoretical insights can be extended to different material systems. Considering the potential impacts the current work will lead to, I recommend it for publication after the authors address my following concerns.

Response: We sincerely thank the reviewer for recognizing the novelty and results of our work. In response to the feedback provided, we have meticulously prepared a point-by-point reply and revised the manuscript, accordingly, addressing each of the specific questions and suggestions raised. We are confident that these revisions significantly enhance the clarity and detail of our manuscript.

1. The logic links between the three parts of the contents should be strengthened. For example, the effects of AS structures, and the grain boundaries on the nucleation and growth of 2D materials, although maybe reported in previous studies, should be included for completeness.

Response: We appreciate the important comment from the reviewer. The nucleation of 2D vdW layered materials on an atomic sawtooth surface and the single-crystal growth through seamless stitching of aligned grains have already been reported in previous studies. To provide a logical explanation, we have added a description of these results in the revised manuscript.

- In line 1 of page 8: “It has been observed that 2D grains grown on the AS surface are coherently aligned with each other and these aligned 2D grains coalesce seamlessly, resulting in the formation of large-area SC 2D films with the presence of grain boundaries¹¹⁻²³.”

2. What controls the atomic-level structures of the AS Cu surfaces?

Response: The solidification of molten metal undergoes a process analogous to material growth, encompassing both nucleation and growth phases. The atomic-level structure of the AS Cu surface is predominantly determined by the facets of W where the initial nuclei form during the solidification process. To clarify this issue, we modified a sentence in the revised manuscript.

- In line 29 of page 7: “This experimental observation aligns with our simulation results. While polycrystalline Cu forms near the polycrystalline W surface, the topmost Cu layers are observed to be single crystalline. This phenomenon can be attributed to a predominant Cu

layer nucleating at a specific W facet and then propagating to the entire W grains. Consequently, this process leads to the formation of SC Cu films (Fig. 3c).”

3. The essence of single crystallinity in h-BN on the oxidation-barrier function lacks explanation from, for example, atomistic simulations.

Response: We thank for the reviewer’s an important comment to improve our manuscript. In previous report, hBN films have been proved as the superior barrier for atoms and molecules [references 10, 28-30 in the revised manuscript]. To address the reviewer’s comment, we have added some sentences in the revised manuscript.

- In line 14 of page 8: “Unlike PC hBN, which allows the permeation of gas molecules through its defective grain boundaries, SC hBN films have been demonstrated to be an ideal gas barrier^{10, 28–30}. Hence, the SC hBN film grown on AS Cu surface was further evaluated as oxidation barrier for Cu surfaces.”

4. The thickness dependence of single crystallinity is very interesting. Can the critical value of 90 micrometers be quantitatively estimated from theory since the molecular simulations are limited to nanometer thicknesses? Why the topmost layers are single crystalline (line 188)?

Response: We appreciate your interest in our model and your comments. It is difficult to perform the MD simulations with micro-scale systems because of the time complexity of the simulations. 90 micrometers thickness obtained from the experimental results is hard to be directly connected with our model. Please acknowledge this difficulty.

As the reviewer recognizes, that topmost layer is single crystalline. Based on both MD simulations and experimental observations, a Cu layer nucleates at a specific W facet, where has higher interface energy, and then entirely covers the surface. Eventually, single crystalline Cu layer is formed. Please, refer to our model as shown in Fig. 3c.

To address this issue, we added sentences in the revised manuscript.

- In line 29 of page 7: “This experimental observation aligns with our simulation results. While polycrystalline Cu forms near the polycrystalline W surface, the topmost Cu layers are observed to be single crystalline. This phenomenon can be attributed to a predominant Cu layer nucleating at a specific W facet and then entirely covering the Cu and W grains. Consequently, this process leads to the formation of SC Cu films (Fig. 3c).”

5. Crystal growth (of the metal layers) is a kinetic process. The molecular dynamics simulations were performed at the scale of nanometers, which may be sufficient for nanometer-thick layers but for sure are not enough for the micrometer ones. The authors should change the experimental processing conditions such as the temperature profiles to check the kinetic effects beyond the thermodynamic arguments in Eqs. 1-4.

Response: We agree with reviewer's comments that the crystal growth is a kinetic process. To substantiate this, we further conducted the melting-solidification (MS) experiments with faster cooling rates of 3.3 and 16 °C/sec, in contrast to the optimal cooling rate of 0.2 °C/min, while keeping other conditions unchanged. The results, as depicted in Figure S3 below, show the formation of a polycrystalline Cu film. This is likely due to the simultaneous formation of multiple nuclei during the rapid cooling process. To emphasize the significance of the cooling rate, we have included additional sentences in the revised manuscript and supplementary data in the supporting information.

- In line 2 of page 5: “It is noted that while the optimal cooling rate (< 0.2 °C/min) resulted in the formation of a SC AS metal film, faster cooling rates (3.3 and 16 °C/sec) led to the formation of PC AS metal film (Fig. S3). This observation may be attributed to the initiation of several nuclei simultaneously at multiple points during the rapid cooling process.”

Figure S3 | EBSD images with IPF maps of Cu surfaces after MS process with different cooling rates; a, fast cooling (-3.3 °C/s) and b, quenching (-16 °C/s) at two different regions.

6. There is a lot of information missing from the simulations. For example, why W(255) and W(001) surfaces were chosen?

Response: We appreciate valuable comments about our simulation results. In our original manuscript, we considered several W indices including W(346)/W(557), W(567)/W(227), W(447)/W(156), W(477)/W(112) as shown in Figure S6. Therefore, we chose the W(255)/W(001) result as a representative data. To support those results, we further simulated MS process on W(255)/W(001)/W(135), W(477)/W(112)/W(235), W(567)/W(227)/W(156) as shown in Figure S8.

All data consistently show that the effective thickness (h_{eff}) (or total energy difference ΔE) is critical parameter for the formation of the single crystal Cu film. Therefore, we added this result as Figure S8 in the revised manuscript.

- In line 7 of page 7: “Similar trends are also observed for different W facets (Figs. S8 and S9).”

Figure S9 | MD simulation results of Cu/W system with different three W indices. Simulated results about different three W indices with **a, thin Cu and **b**, thick Cu.**

7. Why can the oxidation layers of Cu and W be neglected?

Response: We appreciate reviewer’s insightful question. The experiments involving melting-solidification process were conducted under a hydrogen atmosphere, which effectively removes the oxidation layers. Therefore, we have neglected the presence of the oxide layers in this study.

Overall, we appreciate the thoughtful comments and valuable suggestions from the reviewer. We have made a necessary revision to fully address the concerns raised by the reviewers, which greatly helped us to improve the manuscript. We believe that our study represents an important advancement in the related field thus it could make a valuable contribution to *Nature Communications*.

Reviewer #4 (Remarks to the Author):

This manuscript elucidates the principles behind the formation of atomic sawtooth metal film for the growth of single-crystal van der Waals (vdW) layered materials. Molecular dynamics simulations and experimental results were employed to investigate the formation mechanism of atomic sawtooth surface during the solidification process of molten metal on polycrystalline metal substrates. The findings underscore the importance of having a sufficient thickness of molten metal for the formation of atomic sawtooth film. Additionally, the authors demonstrated the growth and application of single-crystal vdW layered materials. Given that single-crystal growth of vdW layered materials is a crucial topic in the community, this manuscript introduces novelty suitable for publication in Nature Communications. The manuscript is logically well-organized, and the authors used appropriate research strategies to understand the mechanism. Therefore, I recommend publishing this manuscript in Nature Communications after minor revisions. I have provided some comments below to enhance the completeness of this work.

Response: We thank the reviewer for thoroughly reading our manuscript and providing valuable comments. In response, we have prepared a point-by-point response and revised the manuscript to address each of your insightful questions and suggestions. These revisions have significantly improved the quality of our manuscript.

1. The authors mentioned that “The epitaxial solidification of the molten Cu layer nucleated in high-index W grains and easily extended to other W grains with lower interface energies. This process resulted in a random Miller index of the SC AS Cu film on the PC W substrates owing to the inherent randomness of the W facets.” This seems to be a bit inconsistent. Since the nucleation starts at a high-index surface and extends to form a single crystal, the Cu surfaces cannot be completely random; they should also prefer high-indices. Also, I wonder if there exists a more quantitative relation between the surface indices of W and Cu? e.g. if the highest index surface of the PC W is (211), would it result in (211) surface for SC Cu as well? What is the Cu surface index observed in the MD simulation?

Response: We concur with reviewer’s comment regarding the preferred Cu index. The Cu index is determined by the W facets where the molten Cu initially begins to solidify, as shown in Fig. S8-S11 below. The term ‘random Miller index’ refers to the random nucleation of the Cu layer on W facets, stemming from the inherent randomness of the polycrystalline W facets. To clarify this and avoid ambiguity, we have revised the relevant sentences in the revised manuscript and included Figs. S8-S11 in the revised supplementary information.

- In line 5 of page 9: “The epitaxial solidification of the molten Cu layer nucleated in high-

index W grains and easily extended to other W grains with lower interface energies. This process resulted in a random Miller index of the SC AS Cu film on the PC W substrates. This randomness is attributed to the inherent variability of the W facets, where the Cu layer initially nucleated on the polycrystalline W facets.”

Figure S8 | **a**, MD simulation results for the Cu/W system on the W(346) and W(557) grains with different Cu thicknesses (h_{eff} : effective thickness of Cu). **b**, MD simulation results for other Cu/W systems with various W grains.

Figure S9 | MD simulation results of Cu/W system with different three W indices. Simulated results about different three W indices with **a**, thin Cu and **b**, thick Cu.

Figure S10 | MD simulation results for **a**, Cu/W and **b**, Au/W systems with various Cu and Au thicknesses (h_{eff} : effective height of Cu and Au).

Figure S11 | MD simulation results for **a**, Cu/W and **b**, Au/W systems with various areal ratios of W(255) to W(001) grains (d : grain size, A : surface area).

2. The author performed hBN growth at temperatures exceeding 1000 °C. I am curious whether the index of the atomic sawtooth Cu film remains unchanged at high temperatures.

Response: We sincerely appreciate the reviewer's insightful concern. To address this issue, we performed EBSD measurements on the atomic sawtooth Cu film both before and after the thermal

treatment at high temperature (1050 °C) for 3 h. As shown in Figure R2, an almost identical color contrast was observed before and after the annealing treatment. This indicates that the structure of atomic sawtooth surface remains stable during the growth of hBN.

Figure R2 | EBSD images of the SC AS Cu film. a, before and b, after annealing at 1050 °C for 3 h.

3. Would the cooling rate during the solidification process be an important factor?

Response: The solidifications process involves thermodynamics and kinetics. To better understand the kinetics of the solidification, we conducted additional melting-solidification experiments with faster cooling rates. While our optimum cooling rate (<0.2 °C/min) resulted in the formation of SC AS metal film, faster cooling rates (3.3 and 16 °C/sec) produced polycrystalline AS film as shown in Figure S3. This phenomenon may be attributed to the formation of multiple nuclei, which are initiated simultaneously at various points during the rapid cooling process. Accordingly, we have included these results in the revised supplementary information and discussed their implications in the revised manuscript.

- In line 2 of page 5: “It is noted that while the optimal cooling rate (< 0.2 °C/min) resulted in the formation of a SC AS metal film, faster cooling rates (3.3 and 16 °C/sec) led to the formation of PC AS metal film (Fig. S3). This observation may be attributed to the initiation of several nuclei simultaneously at multiple points during the rapid cooling process.”

Figure S3 | EBSD images with IPF maps of Cu surfaces after MS process with different cooling rates; a, fast cooling (-3.3 °C/s) and b, quenching (-16 °C/s) at two different regions.

Overall, we appreciate the thoughtful comments and valuable suggestions from the reviewer. We have made a necessary revision to fully address the concerns raised by the reviewers, which greatly helped us to improve the manuscript. We believe that our study represents an important advancement in the related field thus it could make a valuable contribution to *Nature Communications*.

REVIEWER COMMENTS

Reviewer #1 (Remarks to the Author):

The revision has some helpful improvements. Yet the Reviewers, especially those aware of prior growth on stepwise surfaces [11-15], had valid general reservations on the degree of novelty expected for a cross-disciplinary (!) journal as Nat Comm. (E.g. R2 "is not well convinced with the novelty".) My opinion, it still may become passable, but in such case the emphasis must be at least on clarity and accuracy. At the very least, two points I find still fail these criteria.

*** Former 1. The distinction between standard term vicinal and a cute neologism "sawtooth" appears immaterial, and Fig R1 is incomprehensible: (b) appears same, merely rotated compared to (a). Ref to authors' own prior use in Adv. Mater. (2021) is of no help; the only incentive to retain this term seems that by repeating it few times they will eventually claim that have "discovered/introduced" it – but it is essentially the same, and new name can only misleads readers and clutters scientific narrative. "We said it before, so we also can say it here" is an abuse of mathematical induction principle :-) Just call it vicinal, keep clean.

*** Former 7. I stand by my objection (7.) and recommend removing this block, all that chatter about "subgroups", lines 52-55. Ref. to prior Nat Com [23] is unhelpful (same fallacy of "math-induction" in action: someone said it so we also can say) I have no time and it is out of place to discuss its merits, but there then "trivial subgroup of any group is the subgroup {e} consisting of just the identity element", so the amorphous no-symmetry substrate would serve best? which is absurd. Remove it.

With these two above clear and easy, but mandatory revisions paper may become acceptable.

Reviewer #2 (Remarks to the Author):

Thank you for the detailed response and additional experiments and simulation to my concerns. I think that it needs more information on the following two topics before the publication in Nature Communications.

1. Atomic sawtooth (AS) surface structure

Not only me but also the reviewer 1 also expressed the concern about the definition of AS structure and difference from the standard step-terrace structure. As displayed in the attached image, if you tilt the AS surface, the structure is almost identical to that one seen in the step-terrace structure. In addition, the experimental evidence of the AS structure is only Fig. 1g, and the observed area is extremely small (the width is composed of only 10 atoms, except for the artificially highlighted atoms). Therefore, I suggest to present more data to support the authors' claim.

2. >Figure S3. EBSD images with IPF maps of the solidified Cu surfaces with different cooling rates of a, fast cooling 3.3 °C/s) and b, quenching cooling 16 °C/s)."

However, this is not well supported, because they show EBSD images in Figure S3, but there is no evidence of the formation of AS surface for different cooling rates, because there is no AFM/STEM images.

Reviewer #3 (Remarks to the Author):

My previous comments were addressed in the revision and now I recommend acceptance.

Reviewer #4 (Remarks to the Author):

The authors have fully addressed my previous comments

Reviewer #1 (Remarks to the Author):

The authors present a methodology for the production of large-scale single-crystal Cu/Au films with a high-index (“sawtooth”) surface on top of the polycrystalline W/Mo films. Further, they propose the mechanism of the single-crystal formation as shown with MD simulations, and show the effectiveness of the method by using the produced film as a substrate for the growth of the monocrystal hBN.

The use of the vicinal substrate steps to align nuclei of 2D materials was recently shown to be effective, motivating the presented work, where the methodology appears effective and produces good results. However, authors need to elaborate on the importance of their contribution, and the theoretical modeling part has several aspects that must be clarified, corrected, or perhaps deleted not to blur the story. Based on that, I recommend a major revision.

Response: We would like to thank you for your thoughtful and valuable comments to improve our manuscript. We have thoroughly revised the manuscript based on reviewer’s comments by the clarification, correction, and deletion of the suggested parts. We believe that our responses would address all your concerns and that you will find our manuscript suitable for publication.

1. The term “sawtooth surface” is intuitive but uncommon, not defined in the manuscript and is used only once outside of the abstract and title. Vicinal surface is well established in science and technology; No need to introduce a novel term in this context.

Response: We appreciate the reviewer's understanding of the surface structures. As known, a vicinal surface comprises steps and terraces, typically created by mis-cutting a high-symmetry surface at a slight angle, resulting in a 'stair-like' appearance, as illustrated in Fig. R1. In contrast, while an atomic sawtooth structure also includes 'steps and terraces', its appearance is distinctly different, resembling an 'atomic sawtooth' rather than stairs, We hope the reviewer acknowledges this difference, which justifies our use of the term 'atomic sawtooth surface' as described in our previous paper [S. H. Choi et al. Adv. Mater. 33, 2006601 (2021)]. We kindly ask for your understanding in this matter.

Figure R1. Schematics of (a) stair-like vicinal surface and (b) atomic sawtooth surface.

2. As wafer-scale growth of hBN was previously achieved on both vicinal and (111) surfaces (Ref.11-12), and the single crystalline Cu films were also achieved (Ref. 14), authors should elaborate on the importance of their contribution. I note a relevant paper which precedes all Refs 11-14 and must be

cited, K Bets et al. Nano Lett.19, 2027 (2019).

Response: We acknowledge the importance of the references mentioned, which we discussed from page 3, lines 23-33. We would appreciate it if the reviewer could kindly review this section again for clarification. Additionally, we are grateful for the reviewer's suggestion of a relevant paper that was previously not included in our manuscript. In response, we have now added the reference [K. Bets et al., Nano Lett., 19, 2027 (2019)] and incorporated additional sentences in the revised manuscript to reflect this.

- In line 20 of page 3: “The importance of vicinal surfaces for SC growth has also been proposed through studies using density functional theory and molecular dynamics¹¹.”

3. The proposed theoretical model has several significant issues: Why, within the MD simulation, the secondary nucleation does not appear on the W(001) grain within the thicker Cu film? This process should not in any way be affected by the thickness of the film. The detection of the nuclei within the MD simulation is questionable, as a very regular lattice formed by Cu atoms on W(001) grain is visible in every single frame shown, but not highlighted as a nucleus.

Response: We appreciate the valuable comments. We also agree that secondary nucleation occurs irrespective of the thickness of Cu film. However, it was not observed in our MD simulation results for the thick Cu film. This maybe attribute to differences in nucleation times between thin and thick Cu films. Upon examining the system energy changes (as shown in Figures R2a and b), we observed that the first Cu nucleation occurs at a similar time (state II). However, in the thick Cu film, the solidification progresses more rapidly to the neighboring W(001) facet (states III and IV), thereby disrupting secondary nucleation (as shown in Figures R2c and d). Consequently, even at state V, a disordered layer was observed. Given the difficulty in describing this disordered layer as secondary nucleation, we have revised the manuscript to classify it as a buffer layer instead of non-epitaxial layer and have provided an explanation for this issue.

Figure R2 | System energy changes and the simulated results of the MD simulations during the MS process. a,c, thin and b,d, thick Cu films on a PC W surface.

- In line 2 of page 7: “In this state, a disordered interface Cu layer, indicated as buffer layer, is formed between the single crystal Cu layer and W(001) surface. The formation of buffer layers, rather than secondary nucleation, may be attributed to the rapid propagation of the epitaxial Cu layer in thick Cu films, extending to neighboring W(001) surfaces, thereby disrupting secondary nucleation. This rapid propagation is also evident in the system energy changes (Fig. S7).”

Figure S7 | System energy changes of the MD simulations during the MS process for **a**, thin and **b**, thick Cu films on a PC W surface. The states indicate each phase change in Figures 3b and c.

Figure 3 | Theoretical models for the formation of PC and SC AS metal films based on molecular dynamics (MD) simulations during the MS process. **a**, Schematic of two different models: (i) the formation of two different metal grains (A'-index metal and B-index metal) and (ii) development of SC metal (A'-index metal) film formed on a PC W surface (A-index and B-index). The total energy for each model is given by $E_1 = A_A\gamma_e + A_B\gamma_e + A_{GB}\gamma_{GB}$ and $E_2 = A_A\gamma_e + A_B\gamma_{ne}$. The energy difference between the models is $\Delta E = E_1 - E_2 = A_{GB}\gamma_{GB} - A_B(\gamma_{ne} - \gamma_e)$. A and γ denote the area and interface energy of each index and grain boundary, respectively. **b,c**, Depiction of the solidification process from states I to V for **b**, a thin Cu film (i, $\Delta E < 0$) and **c**, a thick Cu film (ii, ΔE

> 0) on W(255) and W(001) surfaces.

4. It is extremely unlikely that the nucleation occurs on the single grain of W within a centimeter-scale polycrystalline film, which is confirmed by the experimental observation of minor grains. This is in no way addressed in the proposed mechanism.

Response: Our experimental data clearly demonstrate the formation of single crystalline Au/Cu metal films on polycrystalline W films. While this result might initially seem extremely unlikely, our works provide robust evidence to the contrary. To elucidate this phenomenon, we developed a simplified model focusing on the formation of metal layers on two distinct types of W grains.

In our molecular dynamics (MD) simulations, it was observed that epitaxial Cu/Au layers predominantly nucleate on high-index W grains. This preference is attributed to the higher interfacial energy associated with these grains. Following nucleation, the Cu/Au layers propagate onto other W grains, ultimately leading to the formation of a single crystalline Cu/Au film over the entire surface, as depicted in Fig. 3c. While nucleation on other W grains is possible, these events are typically overshadowed by the more dominant Cu/Au layer that solidifies earlier on the high-index W grains. This model aligns well with our experimental observations, where the topmost metal layer is single crystalline, whereas the underlying layers are polycrystalline. This concurrence between our model predictions and experimental data underscores the validity of our approach.

We acknowledge the limitation that simulating a system on a centimeter scale is currently unfeasible, primarily due to computational constraints. However, we strongly believe that the scenario observed in our smaller-scale simulations is likely reflective of what occurs in larger systems. Our confidence in this extrapolation is based on the fundamental principles and behaviors observed in our model.

5. The analytics (1-3) represents a thermodynamic equilibrium, not to be achieved within a 13 ns MD simulation. Furthermore, to address the concerns listed above, the $dE > 0$ case should include the secondary grain, GB region, etc.

Response: We agree with the reviewer that MD simulation of 13 ns is significantly shorter than the time scale of real experiments, which is from minutes to hours. However, the sizes of our simulated systems, which contain a few thousand atoms, are also significantly smaller than that of the real experimental materials. Based our experience, such a smaller system can reach thermal equilibrium in a much shorter time scale, such as 100 ps. So, we believe that our simulation time of 13 ns is sufficient for this study. In addition, we have added system energy changes in MD simulation as Fig. S7 and the following explanations in the revised manuscript for secondary nucleation for the $dE > 0$

case.

- In line 2 of page 7: “In this state, a disordered interface Cu layer, indicated as buffer layer, is formed between the single crystal Cu layer and W(001) surface. The formation of buffer layers, rather than secondary nucleation, may be attributed to the rapid propagation of the epitaxial Cu layer in thick Cu films, extending to neighboring W(001) surfaces, thereby disrupting secondary nucleation. This rapid propagation is also evident in the system energy changes (Fig. S7).”

Figure S7 | System energy changes of the MD simulations during the MS process for a, thin and b, thick Cu films on a PC W surface. The states indicate each phase change in Figures 3b and c.

6. In the method section, the sentence “In the MD simulations, the ... (EAM) potentials were used to describe the interactions between the overlayer metal atoms (Cu or Au) and W atoms, and the Morse potential was used to describe the interaction between the overlayer metal atoms and W atoms” fails to specify what Morse and EAM potentials were actually used for.

Response: We appreciate the valuable comments. The EAM potentials were utilized to describe interactions between same metal atoms (such as Cu-Cu, W-W etc.), whereas the Morse potential was employed to represent interactions between different metal atoms (Cu-W). To avoid misleading, we have included additional sentences in the revised manuscript.

- In line 1 of page 12: “In the MD simulations, the embedded atom method was employed to characterize interactions between same metal atoms (such as Cu-Cu, Au-Au, and W-W)^{34,35}, while the Morse potential was utilized to represent interactions between different metal atoms (Cu-W and Au-W)³⁶.”

7. The “...in which the symmetry group belongs to a subgroup of 2D vdW materials” motivation of the substrate surface choice is not convincing, unnecessary, and simply does not belong here, simpler is to remove this block lines 52-55. Following this bizarre logic, the best and most universal substrate would possess no symmetry at all and will work for the growth of all and any 2D material. Furthermore, for most metals at any reasonable growth temperature, the surface is unlikely to maintain the symmetry defined for the idealized structure. So this controversial idea is better to leave out.

Response: As the reviewers are aware, the symmetry relationship between 2D materials and their growth substrate is significantly important. Moreover, it is essential that the growth substrate belongs to a subgroup of symmetry group of the 2D materials [Ref. 23]. This relationship is supported by numerous experimental observations [Refs. 11-19]. But we concur with the reviewer’s comment that the lowest-symmetry group is likely to be represented by the vicinal surface. Consequently, we have retained the sentences in our manuscript that provide guidance for designing the growth substrates and revised the last sentence to more clearly reflect this concept.

- In line 18 of page 3: “According to this concept, the lowest-symmetry substrate, featuring a vicinal surface (C_{1v}) is an ideal growth platform for SC 2D vdW materials²³.”

Overall, we appreciate the thoughtful comments and valuable suggestions from the reviewer. We have made a necessary revision to fully address the concerns raised by the reviewers, which greatly helped us to improve the manuscript. We believe that our study represents an important advancement in the related field thus it could make a valuable contribution to *Nature Communications*.

Reviewer #2 (Remarks to the Author):

This paper reports the mechanism of the formation of Cu and Au single crystal films on polycrystalline W foil. The synthesis of single-crystalline catalyst metals is very important for the growth of high-quality 2D materials, such as hBN, TMD, and graphene. In this manuscript the authors discuss the mechanism of the formation by combining the experimental work (metal thickness dependence (Fig. 4a,b) and the metal etching (Fig. 4c,d)) and theoretical analysis (Fig. 3). I think that the model proposed in Fig 3. is reasonable and well explains the single crystal metal formation by epitaxial solidification.

Response: We are grateful to the reviewer for thoroughly reviewing our manuscript and providing valuable comments. In response, we have prepared detailed point-by-point responses and revised the manuscript to address each of your insightful questions and suggestions. These revisions have significantly enhanced the quality of our manuscript.

1. However, in terms of the aligned growth of 2D materials, demonstrated in Figs. 5 (hBN) and Fig. S12 (graphene and WS₂), the presence of atomic sawtooth (AS) surface structure is essential, as demonstrated in the authors' previous paper (refs. 17,18). However, the present manuscript discusses mainly the formation of the single-crystalline metal films on a polycrystalline W foil. Therefore, the reviewer is not well convinced with the novelty and importance of this work.

Response: Previous reports [18,19] have established the synthesis mechanisms of single-crystal 2D vdW materials on atomic sawtooth (AS) surfaces. However, the formation process of these AS surfaces themselves have largely unexplored. Our work is specifically focused on revealing the underlying mechanism responsible for the formation of a single-crystal, atomic sawtooth (AS) metal films on polycrystalline W films during the melting-solidification (MS) process. Furthermore, we have extended this research to apply the AS Cu film, produced via the MS process, in the growth of single-crystal monolayer hBN film, which has not been previously reported. Additionally, our study demonstrates the use of single-crystal hBN film as an effective oxidation barrier, further highlighting the practical implications of our finding. We believe that the insights gained from this research provide a foundation groundwork for future studies aimed at controlling and exploiting atomic sawtooth surfaces. We hope that this explanation adequately clarifies the novelty and importance of our work, addressing the concerns of the reviewer.

2. At least, the authors should discuss the mechanism of formation of atomic sawtooth (AS) surface structure in the single-crystal metal surface.

Response: We appreciate the valuable comments for our study. The solidification process of molten metal undergoes stages analogous to the growth of materials including nucleation and growth. Therefore, the atomic-level structure of the AS Cu surface is mainly determined by the facets of W where the initial nuclei are formed during the solidification process. Therefore, we modified a sentence in the revised manuscript to address this point.

- In line 29 of page 7: “This experimental observation aligns with our simulation results. While polycrystalline Cu forms near the polycrystalline W surface, the topmost Cu layers are observed to be single crystalline. This phenomenon can be attributed to a predominant Cu layer nucleating at a specific W facet and then entirely covering the Cu and W grains. Consequently, this process leads to the formation of SC Cu films as illustrated in Fig. 3c.”

3. Difference between the well-known step-terrace structure and the proposed AS structure is also unclear for the reviewer.

Response: We appreciate the reviewer's understanding of the surface structures. As known, a vicinal surface comprises steps and terraces, typically created by mis-cutting a high-symmetry surface at a slight angle, resulting in a 'stair-like' appearance. However, while an atomic sawtooth structure also includes 'steps and terraces,' its appearance is distinctly different, resembling an 'atomic sawtooth' rather than stairs, as illustrated in Fig. R1. We hope the reviewer acknowledges this difference, which justifies our use of the term 'atomic sawtooth surface' as described in our previous paper [S. H. Choi et al. Adv. Mater. 33, 2006601 (2021)]. We kindly ask for your understanding in this matter.

Figure R1 | Schematics of (a) stair-like vicinal surface and (b) atomic sawtooth surface.

4. Overall, the reviewer finds it a borderline case for publication in Nature Communications with due to the high scientific standards of the journal. At least it needs more experimental and theoretical explanations on the mechanism of AS structure formation, which is an essential point of the high-quality 2D materials growth.

Response: We appreciate the reviewer's valuable interest on our study. Following reviewer's suggestion, we additionally performed the melting-solidification (MS) experiments with faster cooling

rates to offer the information of kinetic process of crystal growth as the experimental results for the mechanism of AS structure formation. Moreover, we also added the simulation results with three different W facets to support our mechanism and to support our studies as theoretical results. Thus, these have included the following experimental and theoretical results:

1. **Experimental results:** It is observed that as the cooling rate increased during the solidification process of molten Cu, more Cu grains were nucleated (as shown in Fig. S3 below). This aligns with the MD simulation results, which illustrate the formation of a single Cu grain initiating solidification, subsequently covering other areas before another nucleation occurs, thus resulting in a SC AS Cu film. Therefore, we added this result as Fig. S3 and discussed it in the revised manuscript.

- In line 2 of page 5: “It is noted that while the optimal cooling rate (< 0.2 °C/min) resulted in the formation of a SC AS metal film, faster cooling rates (3.3 and 16 °C/sec) led to the formation of PC AS metal film (Fig. S3). This observation may be attributed to the initiation of several nuclei simultaneously at multiple points during the rapid cooling process.”

Figure S3 | EBSD images with IPF maps of the solidified Cu surfaces with different cooling rates of **a**, fast cooling (-3.3 °C/s) and **b**, quenching cooling (-16 °C/s).

2. **Simulation results:** Even when three W facets are present, epitaxial Cu film formation was initially observed on the facet where nucleation occurred. This finding further validates the reliability of our simulation results previously obtained with two W facets. Therefore, we also added this result as Fig. S8 to support our proposed mechanism.

- In line 7 of page 7: “Similar trends are also observed for different W facets (Figs. S8 and S9).”

Figure S9 | MD simulation results of Cu/W system with different three W indices. Simulated results about different three W indices with **a, thin Cu and **b**, thick Cu.**

Overall, we appreciate the thoughtful comments and valuable suggestions from the reviewer. We have made a necessary revision to fully address the concerns raised by the reviewers, which greatly helped us to improve the manuscript. We believe that our study represents an important advancement in the related field thus it could make a valuable contribution to *Nature Communications*.

Reviewer #3 (Remarks to the Author):

This paper presents a combined experimental and theoretical study on the growth of single- and polycrystalline metals, growing 2D materials on top of the metals, and the oxidation barriers endowed by the grown h-BN on Cu. Energy analysis arguments and molecular dynamics simulations explained the experimental findings. The work demonstrates a novel approach to control the texture of metal layers on top of a metal substrate, and the theoretical insights can be extended to different material systems. Considering the potential impacts the current work will lead to, I recommend it for publication after the authors address my following concerns.

Response: We sincerely thank the reviewer for recognizing the novelty and results of our work. In response to the feedback provided, we have meticulously prepared a point-by-point reply and revised the manuscript, accordingly, addressing each of the specific questions and suggestions raised. We are confident that these revisions significantly enhance the clarity and detail of our manuscript.

1. The logic links between the three parts of the contents should be strengthened. For example, the effects of AS structures, and the grain boundaries on the nucleation and growth of 2D materials, although maybe reported in previous studies, should be included for completeness.

Response: We appreciate the important comment from the reviewer. The nucleation of 2D vdW layered materials on an atomic sawtooth surface and the single-crystal growth through seamless stitching of aligned grains have already been reported in previous studies. To provide a logical explanation, we have added a description of these results in the revised manuscript.

- In line 1 of page 8: “It has been observed that 2D grains grown on the AS surface are coherently aligned with each other and these aligned 2D grains coalesce seamlessly, resulting in the formation of large-area SC 2D films with the presence of grain boundaries¹¹⁻²³.”

2. What controls the atomic-level structures of the AS Cu surfaces?

Response: The solidification of molten metal undergoes a process analogous to material growth, encompassing both nucleation and growth phases. The atomic-level structure of the AS Cu surface is predominantly determined by the facets of W where the initial nuclei form during the solidification process. To clarify this issue, we modified a sentence in the revised manuscript.

- In line 29 of page 7: “This experimental observation aligns with our simulation results. While polycrystalline Cu forms near the polycrystalline W surface, the topmost Cu layers are observed to be single crystalline. This phenomenon can be attributed to a predominant Cu

layer nucleating at a specific W facet and then propagating to the entire W grains. Consequently, this process leads to the formation of SC Cu films (Fig. 3c).”

3. The essence of single crystallinity in h-BN on the oxidation-barrier function lacks explanation from, for example, atomistic simulations.

Response: We thank for the reviewer’s an important comment to improve our manuscript. In previous report, hBN films have been proved as the superior barrier for atoms and molecules [references 10, 28-30 in the revised manuscript]. To address the reviewer’s comment, we have added some sentences in the revised manuscript.

- In line 14 of page 8: “Unlike PC hBN, which allows the permeation of gas molecules through its defective grain boundaries, SC hBN films have been demonstrated to be an ideal gas barrier^{10, 28–30}. Hence, the SC hBN film grown on AS Cu surface was further evaluated as oxidation barrier for Cu surfaces.”

4. The thickness dependence of single crystallinity is very interesting. Can the critical value of 90 micrometers be quantitatively estimated from theory since the molecular simulations are limited to nanometer thicknesses? Why the topmost layers are single crystalline (line 188)?

Response: We appreciate your interest in our model and your comments. It is difficult to perform the MD simulations with micro-scale systems because of the time complexity of the simulations. 90 micrometers thickness obtained from the experimental results is hard to be directly connected with our model. Please acknowledge this difficulty.

As the reviewer recognizes, that topmost layer is single crystalline. Based on both MD simulations and experimental observations, a Cu layer nucleates at a specific W facet, where has higher interface energy, and then entirely covers the surface. Eventually, single crystalline Cu layer is formed. Please, refer to our model as shown in Fig. 3c.

To address this issue, we added sentences in the revised manuscript.

- In line 29 of page 7: “This experimental observation aligns with our simulation results. While polycrystalline Cu forms near the polycrystalline W surface, the topmost Cu layers are observed to be single crystalline. This phenomenon can be attributed to a predominant Cu layer nucleating at a specific W facet and then entirely covering the Cu and W grains. Consequently, this process leads to the formation of SC Cu films (Fig. 3c).”

5. Crystal growth (of the metal layers) is a kinetic process. The molecular dynamics simulations were performed at the scale of nanometers, which may be sufficient for nanometer-thick layers but for sure are not enough for the micrometer ones. The authors should change the experimental processing conditions such as the temperature profiles to check the kinetic effects beyond the thermodynamic arguments in Eqs. 1-4.

Response: We agree with reviewer's comments that the crystal growth is a kinetic process. To substantiate this, we further conducted the melting-solidification (MS) experiments with faster cooling rates of 3.3 and 16 °C/sec, in contrast to the optimal cooling rate of 0.2 °C/min, while keeping other conditions unchanged. The results, as depicted in Figure S3 below, show the formation of a polycrystalline Cu film. This is likely due to the simultaneous formation of multiple nuclei during the rapid cooling process. To emphasize the significance of the cooling rate, we have included additional sentences in the revised manuscript and supplementary data in the supporting information.

- In line 2 of page 5: "It is noted that while the optimal cooling rate (< 0.2 °C/min) resulted in the formation of a SC AS metal film, faster cooling rates (3.3 and 16 °C/sec) led to the formation of PC AS metal film (Fig. S3). This observation may be attributed to the initiation of several nuclei simultaneously at multiple points during the rapid cooling process."

Figure S3 | EBSD images with IPF maps of Cu surfaces after MS process with different cooling rates; a, fast cooling (-3.3 °C/s) and b, quenching (-16 °C/s) at two different regions.

6. There is a lot of information missing from the simulations. For example, why W(255) and W(001) surfaces were chosen?

Response: We appreciate valuable comments about our simulation results. In our original manuscript, we considered several W indices including W(346)/W(557), W(567)/W(227), W(447)/W(156), W(477)/W(112) as shown in Figure S6. Therefore, we chose the W(255)/W(001) result as a representative data. To support those results, we further simulated MS process on W(255)/W(001)/W(135), W(477)/W(112)/W(235), W(567)/W(227)/W(156) as shown in Figure S8.

All data consistently show that the effective thickness (h_{eff}) (or total energy difference ΔE) is critical parameter for the formation of the single crystal Cu film. Therefore, we added this result as Figure S8 in the revised manuscript.

- In line 7 of page 7: “Similar trends are also observed for different W facets (Figs. S8 and S9).”

Figure S9 | MD simulation results of Cu/W system with different three W indices. Simulated results about different three W indices with **a, thin Cu and **b**, thick Cu.**

7. Why can the oxidation layers of Cu and W be neglected?

Response: We appreciate reviewer’s insightful question. The experiments involving melting-solidification process were conducted under a hydrogen atmosphere, which effectively removes the oxidation layers. Therefore, we have neglected the presence of the oxide layers in this study.

Overall, we appreciate the thoughtful comments and valuable suggestions from the reviewer. We have made a necessary revision to fully address the concerns raised by the reviewers, which greatly helped us to improve the manuscript. We believe that our study represents an important advancement in the related field thus it could make a valuable contribution to *Nature Communications*.

Reviewer #4 (Remarks to the Author):

This manuscript elucidates the principles behind the formation of atomic sawtooth metal film for the growth of single-crystal van der Waals (vdW) layered materials. Molecular dynamics simulations and experimental results were employed to investigate the formation mechanism of atomic sawtooth surface during the solidification process of molten metal on polycrystalline metal substrates. The findings underscore the importance of having a sufficient thickness of molten metal for the formation of atomic sawtooth film. Additionally, the authors demonstrated the growth and application of single-crystal vdW layered materials. Given that single-crystal growth of vdW layered materials is a crucial topic in the community, this manuscript introduces novelty suitable for publication in Nature Communications. The manuscript is logically well-organized, and the authors used appropriate research strategies to understand the mechanism. Therefore, I recommend publishing this manuscript in Nature Communications after minor revisions. I have provided some comments below to enhance the completeness of this work.

Response: We thank the reviewer for thoroughly reading our manuscript and providing valuable comments. In response, we have prepared a point-by-point response and revised the manuscript to address each of your insightful questions and suggestions. These revisions have significantly improved the quality of our manuscript.

1. The authors mentioned that “The epitaxial solidification of the molten Cu layer nucleated in high-index W grains and easily extended to other W grains with lower interface energies. This process resulted in a random Miller index of the SC AS Cu film on the PC W substrates owing to the inherent randomness of the W facets.” This seems to be a bit inconsistent. Since the nucleation starts at a high-index surface and extends to form a single crystal, the Cu surfaces cannot be completely random; they should also prefer high-indices. Also, I wonder if there exists a more quantitative relation between the surface indices of W and Cu? e.g. if the highest index surface of the PC W is (211), would it result in (211) surface for SC Cu as well? What is the Cu surface index observed in the MD simulation?

Response: We concur with reviewer’s comment regarding the preferred Cu index. The Cu index is determined by the W facets where the molten Cu initially begins to solidify, as shown in Fig. S8-S11 below. The term ‘random Miller index’ refers to the random nucleation of the Cu layer on W facets, stemming from the inherent randomness of the polycrystalline W facets. To clarify this and avoid ambiguity, we have revised the relevant sentences in the revised manuscript and included Figs. S8-S11 in the revised supplementary information.

- In line 5 of page 9: “The epitaxial solidification of the molten Cu layer nucleated in high-

index W grains and easily extended to other W grains with lower interface energies. This process resulted in a random Miller index of the SC AS Cu film on the PC W substrates. This randomness is attributed to the inherent variability of the W facets, where the Cu layer initially nucleated on the polycrystalline W facets.”

Figure S8 | **a**, MD simulation results for the Cu/W system on the W(346) and W(557) grains with different Cu thicknesses (h_{eff} : effective thickness of Cu). **b**, MD simulation results for other Cu/W systems with various W grains.

Figure S9 | MD simulation results of Cu/W system with different three W indices. Simulated results about different three W indices with **a**, thin Cu and **b**, thick Cu.

Figure S10 | MD simulation results for **a**, Cu/W and **b**, Au/W systems with various Cu and Au thicknesses (h_{eff} : effective height of Cu and Au).

Figure S11 | MD simulation results for **a**, Cu/W and **b**, Au/W systems with various areal ratios of W(255) to W(001) grains (d : grain size, A : surface area).

2. The author performed hBN growth at temperatures exceeding 1000 °C. I am curious whether the index of the atomic sawtooth Cu film remains unchanged at high temperatures.

Response: We sincerely appreciate the reviewer's insightful concern. To address this issue, we performed EBSD measurements on the atomic sawtooth Cu film both before and after the thermal

treatment at high temperature (1050 °C) for 3 h. As shown in Figure R2, an almost identical color contrast was observed before and after the annealing treatment. This indicates that the structure of atomic sawtooth surface remains stable during the growth of hBN.

Figure R2 | EBSD images of the SC AS Cu film. a, before and b, after annealing at 1050 °C for 3 h.

3. Would the cooling rate during the solidification process be an important factor?

Response: The solidifications process involves thermodynamics and kinetics. To better understand the kinetics of the solidification, we conducted additional melting-solidification experiments with faster cooling rates. While our optimum cooling rate (<0.2 °C/min) resulted in the formation of SC AS metal film, faster cooling rates (3.3 and 16 °C/sec) produced polycrystalline AS film as shown in Figure S3. This phenomenon may be attributed to the formation of multiple nuclei, which are initiated simultaneously at various points during the rapid cooling process. Accordingly, we have included these results in the revised supplementary information and discussed their implications in the revised manuscript.

- In line 2 of page 5: “It is noted that while the optimal cooling rate (< 0.2 °C/min) resulted in the formation of a SC AS metal film, faster cooling rates (3.3 and 16 °C/sec) led to the formation of PC AS metal film (Fig. S3). This observation may be attributed to the initiation of several nuclei simultaneously at multiple points during the rapid cooling process.”

Figure S3 | EBSD images with IPF maps of Cu surfaces after MS process with different cooling rates; a, fast cooling (-3.3 °C/s) and b, quenching (-16 °C/s) at two different regions.

Overall, we appreciate the thoughtful comments and valuable suggestions from the reviewer. We have made a necessary revision to fully address the concerns raised by the reviewers, which greatly helped us to improve the manuscript. We believe that our study represents an important advancement in the related field thus it could make a valuable contribution to *Nature Communications*.

REVIEWERS' COMMENTS

Reviewer #1 (Remarks to the Author):

My Comment 1/ Former 1: I appreciate the authors agree and now removed any mentioning of the questionable "group-subgroup... and symmetry matching", which was not relevant to this report anyway.

Comment2 / Former 7: The authors' argument for neologism "sawtooth" remains, forgive me, silly. Absence of complete periodicity in their observed surface structure (the size of terraces and angles fluctuate a bit) is totally understandable. Perfect periodicity is, of course, unattainable in reality for any surface: low-index (100) would have some extra atoms and island, but no need to give it special name; vicinal surface with sparse (or even dense) steps here sill also have variation in terrace size and curved step-edges. So, the argument for a new name doesn't hold. They themselves provide 3 terms that already account for such deviations (corrugated stepped terrace surface, corrugated vicinal surfaces, or corrugated high-index surface) but still insist on sawtooth as "more intuitive." It is not! Indeed, real sawtooth actually is very regular periodic shape (if not then it is a *bad sawtooth* :-). In my opinion, objections from *both* reviewers clearly demonstrate that this term is far from "intuitive" and only brings confusion. I leave it to the Editor. Published paper, especially in decent journal, is a product well-curated (by referees and also editorial staff), so if the Editors feel that unnecessary saw-tooth is cute and does not clutter scientific language (I think it does), then I don't mind to see it published, I don't mind. But better remove it. Reports is better/cleaner without.

Reviewer #2 (Remarks to the Author):

Thank you for the reply and the additional images and sentences. The authors have properly addressed my concerns, so now I recommend the publication of this manuscript.

Reviewer #1 (Remarks to the Author):

The authors' argument for neologism "sawtooth" remains, forgive me, silly. Absence of complete periodicity in their observed surface structure (the size of terraces and angles fluctuate a bit) is totally understandable. Perfect periodicity is, of course, unattainable in reality for any surface: low-index (100) would have some extra atoms and island, but no need to give it special name; vicinal surface with sparse (or even dense) steps here still also have variation in terrace size and curved step-edges. So, the argument for a new name doesn't hold. They themselves provide 3 terms that already account for such deviations (corrugated stepped terrace surface, corrugated vicinal surfaces, or corrugated high-index surface) but still insist on sawtooth as "more intuitive." It is not! Indeed, real sawtooth actually is very regular periodic shape (if not then it is a *bad sawtooth* :-). In my opinion, objections from *both* reviewers clearly demonstrate that this term is far from "intuitive" and only brings confusion. I leave it to the Editor. Published paper, especially in decent journal, is a product well-curated (by referees and also editorial staff), so if the Editors feel that unnecessary saw-tooth is cute and does not clutter scientific language (I think it does), then I don't mind to see it published, I don't mind. But better remove it. Reports is better/cleaner without.

Response: We would like to thank you for your valuable comment to improve our manuscript. We agree with the reviewer's opinion, but we find the concern regarding the term 'atomic sawtooth' causing confusion to be subjective, and there is no definitive answer. It's worth noting that the term 'sawtooth wave' is used to describe a waveform with varying periods and amplitudes. Therefore, we maintain our belief that the term "atomic sawtooth surface" intuitively represents the atomic structure of our substrate surface, and we hope to use this term in our manuscript.